# Integrated Transcriptomic and Proteomic Analyses Revealed the Mechanism of the Osmotic Stress Response in *Lacticaseibacillus rhamnosus* ATCC 53103

**DOI:** 10.3390/foods14173112

**Published:** 2025-09-05

**Authors:** Wei Luo, Xiaona He, Yuxue Chen, Yewen Xu, Yongliang Zhuang, Yangyue Ding, Xuejing Fan

**Affiliations:** 1Faculty of Food Science and Engineering, Kunming University of Science and Technology, Kunming 650500, China; weiluo_05@163.com (W.L.); hexiaona223@163.com (X.H.); 19188775675@163.com (Y.C.); hoshinochi@163.com (Y.X.); kmylzhuang@163.com (Y.Z.); dingyangyue77@163.com (Y.D.); 2Yunnan Technology Innovation Center of Woody Oil, Kunming 650201, China; 3Yunnan Key Laboratory of Plateau Food Advanced Manufacturing, Kunming 650500, China

**Keywords:** *Lacticaseibacillus rhamnosus* ATCC 53103, Osmotic stress, transcriptomics, proteomics, response mechanisms

## Abstract

*Lacticaseibacillus rhamnosus* (*Lbs. rhamnosus*) is renowned for its tolerance to gastric acid and adaptability to bile and alkaline conditions, and is crucial for intestinal health and immune regulation. In this study, integrated transcriptomic and proteomic analyses were employed to elucidate the response mechanisms of *Lbs. rhamnosus* under osmotic stress, induced by exposure to 0.6 M sodium lactate, which elevates environmental osmotic pressure. It was shown that 792 differentially expressed genes and 138 differentially expressed proteins were detected in *Lbs. rhamnosus* ATCC 53103 treated with osmotic stress. The differential regulation of these genes/proteins mainly includes the inhibition of fatty acid metabolism with membrane structural remodeling (downregulation of the acetyl coenzyme A carboxylase family and fatty acid binding protein family expression), dynamic homeostasis of amino acid metabolism (restriction of the synthesis of histidine, cysteine, leucine, etc., and enhancement of the catabolism of lysine, tryptophan, etc.), and survival-oriented reconfiguration of carbohydrate metabolism (gene expression related to the glycolytic pathway increases, while gene expression related to the pentose phosphate pathway decreases). These synergistic alterations in metabolic regulation may facilitate the adaptive response of *Lbs. rhamnosus* ATCC 53103 to osmotic stress. Overall, our findings deepen the current understanding of the stress response mechanisms in lactic acid bacteria and offer novel insights into the survival strategies employed by *Lbs. rhamnosus* ATCC 53103 under hyperosmotic conditions.

## 1. Introduction

Lactic acid bacteria (LAB) are Gram-positive microorganisms that are capable of fermenting carbohydrates, and producing lactic acid as the primary metabolic end product. Due to their unique functional characteristics, LAB have significant applications in food and medicine [1]. *Lbs. rhamnosus* is a species of LAB that is widely distributed in the gastrointestinal tracts of humans and animals. It exhibits a strong tolerance to gastric acid, adapts well to bile and alkaline conditions, and plays a crucial role in maintaining intestinal health and microbial homeostasis [2]. According to the existing research, certain strains of *Lbs. rhamnosus* may alleviate specific types of diarrhea, such as antibiotic-associated diarrhea [3]. Some studies also suggest that these LAB may have positive effects on glucose metabolism and insulin sensitivity [4]. Furthermore, it produces key metabolites, such as short-chain fatty acids (e.g., acetic acid, propionic acid, and butyric acid) and lactic acid, which contribute to maintaining intestinal microbial balance and supply energy to the host intestinal epithelium [5]. Additionally, certain strains of *Lbs. rhamnosus* may play a role in immune regulation and have potential associations with allergic reaction regulation [6].

However, LAB are exposed to various environmental stresses (such as acid, osmotic, temperature, oxidative, bile, etc.) during production, processing, and transportation, which may lead to cellular damage or even cell death. As fermentation progresses, lactic acid and acetic acid produced by LAB accumulate continuously, leading to a progressive decrease in environmental pH and subsequent acidification of the bacterial cytoplasm. This intracellular acidification adversely affects the physiological functions of the cells and significantly impairs their fermentation efficiency [7]. To maintain a stable environment during fermentation, exogenous neutralizing agents (such as sodium hydroxide) are commonly introduced to counteract lactic acid accumulation and reduce its inhibitory effects on LAB. However, over-neutralization results in the accumulation of sodium lactate. This leads to increased osmotic pressure, which causes cellular damage, changes bacterial metabolism, and triggers adaptive responses at the gene and protein levels [8]. For example, under osmotic stress, the genes *GshAB*, *GshR3*, *PepN*, *GshR4*, and *serA*, which are associated with amino acid metabolism, are upregulated in *Lactobacillus plantarum* FS5-5. In addition, I526_2330, *Gpd*, and *Gnd*, which participate in carbohydrate metabolism, also show increased expression. *Lactobacillus plantarum* FS5-5 protects cellular proteins and macromolecules by accumulating compatible solutes, elevating GSH levels, and upregulating DNA repair proteins [9]. Another study found that under osmotic pressure stress, *Lacticaseibacillus paracasei* Zhang transformed into a viable but non-culturable (VBNC) state. Moreover, while maintaining cell integrity, the loss of surface smoothness and the increase in cell aggregation were observed. Further transcriptomic analysis revealed that genes related to carbohydrate and nutrient transport (such as PTS and ABC transport proteins) were upregulated, suggesting that this strain may adapt to osmotic pressure stress and maintain survival by enhancing substance uptake and metabolic capabilities [10]. Additionally, nontargeted metabolomic analysis of *Bifidobacterium bifidum* CCFM16 (*Bb. Bifidum* CCFM16) during osmotic adaptation revealed an upregulation of F6PPK, a key enzyme in the bifid shunt pathway. This finding suggests that the cells redirected energy from the basal metabolism toward the synthesis of osmoprotectants to mitigate osmotic stress. Under prolonged hyperosmotic stress, *Bb. Bifidum* CCFM16 developed a protective mechanism by converting glutamic acid to proline, thereby establishing an osmoprotective system that primarily relies on proline as the adaptive solute [11].

In recent years, advances in high-throughput genomic technologies, coupled with decreasing sequencing costs, have established comprehensive multi-omics analyses as a powerful strategy for elucidating the stress response mechanisms of LAB [12]. However, the survival mechanism of *Lbs. rhamnosus* ATCC 53103 in a hyperosmotic environment is still unknown. To further investigate its response to osmotic stress, the present study examined the mechanism of its resistance at the gene and protein expression levels through the joint analysis of transcriptomics and proteomics techniques. By elucidating the molecular response mechanisms of LAB under osmotic stress, this study provides a theoretical foundation for optimizing production processes in the food industry, including fermented dairy products and pickled vegetables, as well as for developing high-value-added products such as extracellular polysaccharides and probiotic preparations. These findings also offer practical guidance for addressing environmental stress fluctuations and improving the tolerance of bacterial strains.

## 2. Materials and Methods

### 2.1. Bacterial Strains and Culture Conditions

The *Lbs. rhamnosus* ATCC 53103 strain used in this study was preserved at the Faculty of Food Science and Engineering, Kunming University of Science and Technology (Kunming, 650500, China). The preserved strain was inoculated in an MRS (de Man, Rogosa, and Sharpe) broth medium (Qingdao Hope Bio-Technology Co., Ltd., Qingdao, China) at 37 °C for two generations as a seed solution. Previous laboratory studies indicate that a concentration of 0.6 M sodium lactate (Shanghai Macklin Biochemical Co., Ltd., Shanghai, China) represents the sublethal concentration for *Lbs. rhamnosus* ATCC 53103, with late logarithmic growth occurring after 24 h [13]. Therefore, the activated *Lbs. rhamnosus* ATCC 53103 was inoculated with a 2% inoculum at 37 °C in an MRS broth medium containing 0.6 M sodium lactate (Shanghai Macklin Biochemical Co., Ltd.) or without sodium lactate. This was recorded as the SL group (containing sodium lactate, the experimental group) and the MRS group (does not containing sodium lactate, control group). After 24 h, the bacteria cells were harvested for further analysis.

### 2.2. Sample Collection, Pretreatment, and Storage

For the transcriptomics analysis, we collected cells during the logarithmic growth phase, then centrifuged them at 8000× *g* for 5 min at 4 °C. The resulting pellet was frozen in liquid nitrogen and stored at −80 °C until further analysis.

For the proteomics analysis, we collected cells during the logarithmic growth phase. The fermentation broth was collected at the end of the logarithmic growth phase, then centrifuged them (as mentioned above), and the pellet was collected. The pellet was washed three times with phosphate-buffered saline (PBS) (Shanghai yuanye Bio-Technology Co., Ltd., Shanghai, China), frozen in liquid nitrogen, and stored at −80 °C until further analysis.

Three biological replicates were set up for each group. All the samples were frozen and stored for subsequent detection and analysis.

### 2.3. Transcriptomics Analysis by RNA Sequencing (RNA-Seq)

#### 2.3.1. RNA Extraction and Preprocessing

The sample was ground in liquid nitrogen using an appropriate amount. Trizol reagent (Thermo Fisher Scientific, Carlsbad, CA, USA) was added, mixed thoroughly, and incubated for 10 min to ensure complete cell lysis. Chloroform (Guangzhou chemical reagent factory, Guangzhou, China) was added, mixed by inversion, and centrifuged at 14,000× *g* for 10 min at 4 °C. The mixture separated into organic and aqueous phases. The upper aqueous phase was collected, and equal volumes of chloroform and isopropanol (Guangzhou chemical reagent factory) were added. Following centrifugation, the supernatant was discarded, and the pellet was washed with 75 % ethanol (Guangzhou chemical reagent factory), vacuum-dried for 2–4 min, and resuspended in an appropriate volume of RNase free water. The solution was incubated at room temperature for 10 min to ensure complete dissolution, then mixed thoroughly, briefly centrifuged, and stored at −80 °C.

#### 2.3.2. Library Construction, Quality Control, and Sequencing

Total RNA was purified using Agencourt RNA Clean XP Beads (Beckman Coulter, A63987, Beverly, MA, USA) to deplete ribosomal RNA (rRNA). First and second strand cDNA synthesis was performed using a PCR thermal cycler. The resulting double-stranded cDNA was purified using 1.8 × Agencourt AMPure XP Beads (Beckman Coulter), and the supernatant was collected. Adapter ligation and end repair were subsequently performed using a PCR thermal cycler (Dongsheng Xingye Scientific Instruments Co., Ltd., Suzhou, China). The USER enzyme was added, and the mixture was incubated at 37 °C for 15 min. The product was then purified using AMPure XP Beads (Agencourt Bioscience), washed with 80% ethanol, eluted with ddH_2_O, and amplified by PCR, followed by a final purification. Library quality was assessed using the ABI StepOnePlus Real-Time PCR System (Applied Biosystems, Carlsbad, CA, USA). The final library was sequenced on the Illumina NovaSeq X Plus platform (Illumina, San Diego, CA, USA).

#### 2.3.3. Analysis of Differentially Expressed Genes in RNA-Seq Datasets

To compare gene expression levels across genes and samples, the dataset was normalized using the fragments per kilobase of transcript per million mapped reads (FPKM) method [14]. Gene expression levels were estimated using RSEM software (v1.3.3) and normalized using the FPKM method to eliminate the effects of gene length and sequencing depth. Differentially expressed genes (DEGs) were identified using the edgeR package (version 3.12.1), with a fold change ≥ 2 and a false discovery rate (FDR) < 0.05 as the screening threshold. Functional enrichment analysis of DEGs was performed using Gene Ontology (GO) terms and Kyoto Encyclopedia of Genes and Genomes (KEGG) pathways, with a significance threshold of *q*-value < 0.05.

### 2.4. Proteomics

#### 2.4.1. Protein Extraction and Pretreatment

Samples were thawed and lysed in an appropriate volume of lysis buffer containing 1% sodium deoxycholate and 8 M urea (Shanghai Macklin Biochemical Co., Ltd.), along with 1 × protease inhibitor to prevent protease activity. The mixture was vortexed, mixed, and homogenized three times using a high-throughput tissue mill. The resulting lysate was incubated in a sedimentation chamber at 4 °C for 30 min, with vortexing every 10 min. Samples were centrifuged at 14,000× *g* for 20 min at 4 °C. The resulting supernatant was collected, and protein concentration was quantified using the Pierce™ Rapid Gold BCA Protein Assay Kit (Thermo Fisher Scientific, A53225, Waltham, MA, USA). Sample pretreatment involved protein denaturation, reduction and alkylation, enzymatic digestion, and peptide desalting. Protein pretreatment was carried out using the iST Sample Preparation Kit (PreOmics, iST-96-001, Planegg, Germany). An appropriate amount of protein was mixed with 50 µL of lysis solution. The sample was heated at 95 °C for 10 min at 1000× *g*. After cooling to room temperature, trypsin digestion buffer was added, and the sample was incubated at 37 °C for 2 h with shaking at 500× *g*. The enzymatic reaction was terminated by the addition of termination buffer. Peptides were desalted using the iST cartridge and eluted with two 100 µL portions of elution buffer. The eluted peptides were vacuum-dried and stored at −80 °C.

#### 2.4.2. Data Independent Acquisition (DIA) Acquisition

The desalted, lyophilized peptides were redissolved in a solution of phase A (0.1% formic acid in water, Sigma-Aldrich Merck KGaA, Darmstadt, Germany) and analyzed by LC-MS/MS. The complete system consisted of a tandem UltiMate 3000 (Thermo Fisher Scientific, MA, USA) and a timsTOF Pro2 mass spectrometer (Bruker Daltonics). The samples were separated using an AUR3-15075C18 analytical column (15 cm × 75 μm i.d, 1.7 μm particle size, 120 A pore size, IonOpticks) with a 60 min gradient at a column temperature of 50 °C. The column flow rate was controlled at 400 nL/min and the gradient started with 4% of phase B (80% acetonitrile and 0.1% formic acid, Sigma-Aldrich Merck KGaA, Darmstadt, Germany) and rose to 28% in 25 min, 44% in 10 min, 90% in 10 min, maintained for 7 min, and equilibrated at 4% for 8 min. The mass spectrometer was operated in diaPASEF mode to acquire DIA data, with a scanning range of 349–1229 m/z and an isolation window width of 40 Da. During the PASEF MS/MS scan, the collision energy increased linearly with ion mobility, rising from 59 eV (1/K0 = 1.6 Vs/cm^2^) to 20 eV (1/K0 = 0.6 Vs/cm^2^).

#### 2.4.3. Data Analysis

The Data of DIA were processed and analyzed by Spectronaut 18 (Biognosys AG, Schlieren, Switzerland) with default settings. Specific trypsin was set as the digestion type and digestion enzyme. Carbamidomethyl on cysteine was specified as the fixed modification. Oxidation on methionine was specified as the variable modifications. The retention time prediction type was set to dynamic iRT. Data extraction was determined by Spectronaut based on the extensive mass calibration. Spectronaut was to determine the ideal extraction window dynamically depending on iRT calibration and gradient stability. Qvalue (FDR) cutoff on the precursor level was 1% and the protein level was 1%. The decoy generation was set to mutated, which is similar to scrambled but will only apply a random number of AA position swamps (min = 2, max = length/2). The normalization strategy was set to Local normalization. Peptides which passed the 1% Qvalue cut off were used to calculate the major group quantities with the MaxLFQ method.

## 3. Results and Discussion

### 3.1. Transcriptomic Analysis of Lbs. rhamnosus ATCC 53103 Under Osmotic Stress

To elucidate the role of osmotic stress in regulating specific pathways and tolerance mechanisms, a comprehensive transcriptomic analysis was conducted on *Lbs. rhamnosus* ATCC 53103. Differentially expressed genes (DEGs) were identified and compared between the SL and MRS groups to reveal overall transcriptional changes (Figure 1). As shown in Figure 1A, principal component analysis (PCA) demonstrated a high level of consistency among biological replicates of *Lbs. rhamnosus* ATCC 53103 before and after exposure to osmotic stress. Moreover, a clear distinction in gene expression profiles was observed between the SL and MRS groups. The results of the differential clustering analysis are shown in Figure 1B, where the samples within the MRS group and the samples within the SL group were each clustered closely together, indicating a high degree of consistency in gene expression within each group. In contrast, the MRS and SL groups were clearly separated, suggesting that *Lbs. rhamnosus* ATCC 53103 underwent substantial alterations in gene expression patterns following osmotic stress. As illustrated by the volcano plot, RNA-seq analysis identified a total of 792 DEGs, with 547 upregulated and 245 downregulated (Figure 1C and Appendix A).

To better understand the functional impact of osmotic stress on *Lbs. rhamnosus* ATCC 53103, Gene Ontology (GO) enrichment analysis was performed on the DEGs using the GO database (Figure 1D). GO enrichment showed that DEGs were mainly focused on the Biological Process (BP), Molecular Functions (MF), and Cellular Component (CC). In the BP category, the most significantly enriched terms were related to metabolism, cellular processes, localization, and stress responses. Most of the DEGs were associated with fundamental metabolic and cellular maintenance functions, indicating a global adjustment in cellular activity. For the MF category, catalytic activity and binding were the predominant functional classes, suggesting that under osmotic stress, cells may regulate enzyme activity as well as energy utilization and transport mechanisms to preserve homeostasis. In contrast, genes in the CC category were primarily associated with intracellular structures and protein-containing complexes. These findings suggest that the cellular changes induced by osmotic stress primarily affect intracellular structures and may also involve membrane-associated components. Furthermore, the Kyoto Encyclopedia of Genes and Genomes (KEGG) enrichment analysis of DEGs between the SL and MRS groups is shown in Figure 1E. The DEGs were significantly enriched in several key metabolic pathways, including fatty acid metabolism/biosynthesis, ABC transporter system, pyruvate metabolism, and carbohydrate metabolism.

### 3.2. Proteomic Analysis of Lbs. rhamnosus ATCC 53103 Under Osmotic Stress

Proteomics is a powerful platform technology that enables the comprehensive profiling of protein expression patterns [15]. To elucidate the mechanisms by which lactic acid bacteria respond to osmotic stress, a proteomic approach was employed to analyze differentially expressed proteins (DEPs) in the strains before and after exposure to osmotic stress. Principal component analysis (PCA) revealed distinct separation between the protein profiles of *Lbs. rhamnosus* ATCC 53103 before and after osmotic stress, indicating notable differences in protein composition (Figure 2A). Hierarchical clustering analysis of DEPs further demonstrated a marked difference in protein expression abundance between the MRS and SL groups (Figure 2B), suggesting substantial proteomic shifts in response to osmotic stress. Based on the volcano plot, a total of 138 DEPs were identified, comprising 43 upregulated and 95 downregulated proteins (Figure 2C and Appendix A).

To further investigate the functional impact of osmotic stress on *Lbs. rhamnosus* ATCC 53103, DEPs were annotated and subjected to enrichment analysis using the GO database (Figure 2D). The number of DEPs was quantified under the secondary level GO functional categories, and comparisons were made across different GO partitions. The enriched DEPs were classified into 21 BPs, 11 MFs, and 3 CCs. In the BP category, the enriched DEPs were primarily associated with cellular and metabolic processes, as well as localization, biological regulation, and responses to stimuli. Within the MF category, the DEPs were mainly involved in catalytic activity, binding, transporter activity, and ATP-dependent functions. In terms of CC, the differentially abundant proteins were predominantly linked to structural elements and protein complexes. KEGG pathway enrichment analysis further indicated that the DEPs in *Lbs. rhamnosus* ATCC 53103 were mainly involved in amino acid, carboxylic acid, and fatty acid metabolism, as well as glutathione metabolism, in response to osmotic stress (Figure 2E).

### 3.3. Integrated Transcriptomic and Proteomic Analysis of Lbs. rhamnosus ATCC 53103 Gene/Protein Expression Under Osmotic Stress

Bacterial stress regulation involves a complex, multilayered network that spans from gene expression to protein function. Single-omics analyses provide only localized and fragmented insights, making it challenging to comprehensively elucidate regulatory networks and their interactions. This limitation hinders a full understanding of stress response mechanisms. In contrast, multi-omics approaches integrate transcriptomic and proteomic data, enabling a more systematic and holistic analysis of complex regulatory pathways [16]. Integrated transcriptomic and proteomic analyses provide valuable insights into the roles of genes and proteins in the resistance of LAB to osmotic stress. As shown in Figure 3A, there is a significant but weak positive correlation between mRNA and protein expression levels (Pearson’s r = 0.273, *p* < 0.05), indicating that under stress conditions, there is limited coordination between the transcriptional and proteomic responses. Furthermore, as presented in Figure 3B and Appendix A, a total of 73 genes and corresponding proteins exhibited overlapping differential expression following osmotic stress. These included *accA*, *accB*, *fabF*, *fabZ*, *spxB*, *glgA*, *hisF*, and *purC*, among others. A total of 18 genes were found to be upregulated in the third quadrant at both the transcriptional and protein levels. GO enrichment analysis revealed that these genes are primarily involved in key physiological processes such as purine metabolism, nucleotide binding, and transport. These research results suggest that *Lbs. rhamnosus* ATCC 53103 may improve its adaptability to hypertonic environments to a certain extent by regulating energy metabolic pathways (Figure 3C). Consistently, KEGG enrichment analysis revealed that the differentially expressed genes were predominantly associated with pathways related to energy production and nucleotide metabolism, including purine metabolism, the pentose phosphate pathway, glycolysis, and pyruvate metabolism. These findings suggest that *Lbs. rhamnosus* ATCC 53103 may undergo metabolic reprogramming to meet the elevated energy and biosynthetic demands imposed by osmotic stress (Figure 3D). Furthermore, this section focuses on the effects of osmotic stress on fatty acid, amino acid, and carbohydrate metabolism in *Lbs. rhamnosus* ATCC 53103, and elucidates the regulatory pathways that potentially govern these metabolic responses (Figure 4).

#### 3.3.1. Effect of Osmotic Stress on Fatty Acid Metabolism of *Lbs. rhamnosus* ATCC 53103

Fatty acid biosynthesis in cell membranes is considered a crucial process in the cellular response to environmental stress. As shown in Table 1, 11 DEGs related to fatty acid synthesis were downregulated under osmotic stress. These genes are primarily involved in fatty acid biosynthesis and carbon chain elongation. This finding indicates that in a high osmotic pressure environment, cells may inhibit the synthesis of fatty acids to save energy and metabolic resources, which is part of an adaptive regulatory strategy [17]. This observation aligns with a phenomenon frequently reported in lactic acid bacteria. Under external stress conditions, the expression of key genes involved in fatty acid metabolism is often downregulated, suggesting that osmotic stress may directly inhibit the biotransformation and synthesis of fatty acids within bacterial cells. Consequently, this downregulation may affect the intracellular concentrations of organic acids [18]. The only upregulated gene, FG342_RS11585, which encodes acetyl-CoA C-acyltransferase, may be involved in fatty acid degradation or other stress-related metabolic pathways, potentially compensating for the downregulation of the biosynthetic pathway. Proteomics analysis (Figure 5A and Appendix A) revealed that the expression of AdhE, which encodes the bifunctional acetaldehyde-CoA/alcohol dehydrogenase, was upregulated. This might reflect the adaptive response of the cells to the damage induced by osmotic stress or the imbalance of energy metabolism. On the other hand, the expression levels of ACC and FAB family proteins were significantly downregulated. For example, AccA and AccB, which catalyze the conversion of acetyl-CoA to malonyl-CoA and contribute to carbon chain elongation during fatty acid biosynthesis, were markedly reduced. Similarly, the expression of FabZ and FabF, which regulate the introduction of unsaturated double bonds in the carbon chain and influence the relative content of unsaturated fatty acids, was also significantly decreased [19,20]. This result is consistent with gene expression patterns observed in the transcriptomic analysis. Under osmotic stress, cells may reduce energy expenditure or alter membrane lipid composition by actively downregulating the expression of ACC and FAB family proteins, thereby modulating membrane fluidity [21].

#### 3.3.2. Effect of Osmotic Stress on Amino Acid Metabolism of *Lbs. rhamnosus* ATCC 53103

During cellular metabolism, amino acids serve not only as essential precursors for cellular structures but also as active participants in intracellular biochemical reactions and metabolic regulation. They contribute to the formation of catalytic enzymes and enhance the cell’s ability to withstand external environmental stresses [22]. The experimental results indicated that osmotic stress significantly influenced the expression of genes associated with amino acid metabolism (Figure 5B and Table 2). Histidine is a glycogenic amino acid that enters the gluconeogenic pathway under adverse conditions. All seven differentially expressed genes involved in the histidine metabolic pathway were downregulated under osmotic stress, suggesting that histidine may reduce endogenous glucose production by suppressing its own synthesis [23]. In contrast, all seven DEGs involved in the histidine metabolic pathway were downregulated, suggesting that stress may inhibit the de novo synthesis of histidine in *Lbs. rhamnosus* ATCC 53103. In the D-amino acid metabolism pathway, the expression levels of *dltA* and FG342_RS03230 (encoding D-alanine--D-alanine ligase) were significantly upregulated, indicating that this regulatory response may alter the structure and composition of the cell wall and membrane, thereby helping to maintain cellular integrity under osmotic stress [24]. The expression of FG342_RS11585 (encoding acetyl-CoA C-acyltransferase) was upregulated in the lysine degradation, tryptophan metabolism, and valine, leucine, and isoleucine degradation pathways. This gene encodes a key enzyme that converts amino acid carbon backbones into acetyl-CoA, thereby channeling amino acid degradation products into the TCA cycle or fatty acid synthesis pathway. This metabolic regulatory function may help offset the energy and material deficiencies resulting from reduced fatty acid synthesis capacity [25]. Additionally, pyridoxal phosphate-dependent aminotransferase (FG342_RS02445) catalyzes the conversion of amino acids into pyruvate, producing ammonia (NH_3_) as a byproduct. The expression of its encoding gene increased by 2.28-fold in the cysteine and methionine metabolism pathway following osmotic stress. This suggests that the enzyme may regulate the intracellular environment by reducing acidification within cells in the experimental group and modulate energy production to promote cell growth [26]. Proline is a typical osmoprotective agent that contributes to cellular homeostasis by regulating intracellular osmotic pressure. In the proline metabolism pathway, the expression of *proC* (encoding pyrroline-5-carboxylate reductase) is upregulated, thereby promoting proline biosynthesis. Meanwhile, the expression of FG342_RS12380 (encodes a proline-specific peptidase family protein) was downregulated, thereby reducing proline catabolism. These two regulatory mechanisms work synergistically to promote the accumulation of intracellular proline. Notably, the expression of *purQ* was upregulated in pathways related to D-amino acid metabolism, arginine and proline metabolism, and alanine, aspartate, and glutamate metabolism, thereby facilitating the conversion of amino acids into purines and nucleotides and contributing to the maintenance of nucleic acid metabolic homeostasis [27]. As shown in Figure 5B and Appendix A, the protein expression levels of MetE, AdhE, GlyA, HisF, and other proteins associated with amino acid anabolism were significantly upregulated following osmotic stress, as revealed by proteomic analysis. MetE catalyzes the conversion of L-homocysteine to L-methionine, thereby promoting bacterial growth and metabolism, and playing a role in the bacterial stress response [28]. Within the proteomic profile associated with amino acid transport and metabolism, the expression of carbamoyl phosphate synthase (CarB) exhibited a pronounced downregulation. This enzyme plays a pivotal role in the biosynthesis of L-arginine via the ornithine cycle and also serves as a critical component in pyrimidine nucleotide synthesis. Consequently, the downregulation of CarB expression may constrain intracellular arginine biosynthesis and perturb pyrimidine metabolism, thereby influencing associated cellular processes [29]. Integrated transcriptomic and proteomic analyses revealed that *Lbs. rhamnosus* ATCC 53103 mounted an adaptive response encompassing energy production, osmoprotectant accumulation, and efficient resource utilization through the fine-tuned regulation of amino acid metabolism. This metabolic reprogramming may temporarily suppress certain amino acid biosynthetic pathways. However, it may also play a positive role in maintaining essential cellular functions and adapting to hyperosmotic stress.

#### 3.3.3. Effect of Osmotic Stress on Carbohydrate Metabolism of *Lbs. rhamnosus* ATCC 53103

Carbohydrate metabolism is essential for the stress resilience of lactic acid bacteria LAB, as it not only maintains intracellular microenvironmental homeostasis but also plays a critical role in various cellular stress responses [30]. Glycolysis serves as the primary pathway through which microorganisms metabolize sugars to generate energy. This pathway is predominantly driven by the phosphotransferase system (PTS), which facilitates sugar uptake by transporting extracellular sugars into the cell. This pathway is predominantly driven by the phosphotransferase system (PTS), which facilitates sugar uptake by transporting extracellular sugars into the cell. As shown in Figure 5C and Table 3, 6-phospho-beta-glucosidase is a key hydrolase involved in the phosphorylation of disaccharides, typically functioning in conjunction with the PTS to convert carbon sources into glucose-6-phosphate. Under osmotic stress, the expression of the FG342_RS13435 gene, which encodes this enzyme, was upregulated. This upregulation may enhance carbon source utilization and overall metabolic activity [31]. These adaptations may help supply energy and support the maintenance of normal physiological metabolism in the bacterium [32]. Additionally, the expression of PfkB, a protein closely associated with the phosphotransferase system (PTS) and the glycolytic pathway, was upregulated. This upregulation likely promoted increased glucose uptake and facilitated its entry into central metabolism, thereby enhancing both glucose utilization efficiency and ATP synthesis. These combined effects may contribute to improved cellular adaptation of the bacterium under osmotic stress conditions [33]. It is worth noting that L-lactate dehydrogenase is a key enzyme catalyzing the reversible redox conversion between lactate and pyruvate. In the glycolytic pathway, it primarily functions to reduce pyruvate to lactate. The downregulation of its encoding gene, FG342_RS00425, may lead to decreased lactate production. Consequently, more pyruvate may enter gluconeogenesis or the TCA cycle, thereby enhancing the efficiency of energy metabolism [34]. Pyruvate is a key metabolic intermediate that enters the TCA cycle for further oxidation and participates in various metabolic pathways. Upregulation of the gene and protein encoding this enzyme (FG342_RS13400) may enhance acetic acid recovery and reutilization (e.g., entering gluconeogenesis), thereby playing an important role in regulating energy metabolism and improving cellular stress resistance [35]. Notably, the three genes (FG342_RS04910, FG342_RS11565, and FG342_RS13235) encoding pyruvate oxidase and the SpxB protein were downregulated. This suggests a significant inhibition of the metabolic pathway that catalyzes the oxidation of pyruvate to acetic acid. As a result, the metabolic flux of pyruvate is more likely redirected toward alternative pathways, such as the tricarboxylic acid (TCA) cycle or lactic acid fermentation [36]. The pentose phosphate pathway serves as a crucial branch of carbohydrate metabolism, primarily supplying biosynthetic precursors, such as ribulose-5-phosphate, and reducing power in the form of NADPH [37]. As shown in Table 3, genes involved in the pentose phosphate pathway generally exhibited a downward expression trend. Specifically, *gnd* in the oxidative phase, as well as *rpiA* and FG342_RS03995 (encoding phosphopentomutase) in the non-oxidative phase, were downregulated. This suppression may reduce ribulose-5-phosphate synthesis, leading to a disruption in redox balance. Additionally, within the phosphotransferase system (PTS), the expression of both the gene encoding 1-phosphofructokinase (*pfkB*) and the corresponding protein (PfkB) was upregulated. This upregulation may be attributed to the bacterium’s enhanced fructose utilization under osmotic stress and its need for efficient ATP generation to counteract external osmotic pressure [9]. In addition, the expression of the gene (*glgA*) and protein (GlgA), which encode glycogen synthase, was downregulated in starch and sucrose metabolism. This inhibitory effect may promote energy conservation in bacteria by suppressing glycogen synthesis and maintaining ATP levels. This could enhance their ability to adapt to osmotic stress [38]. In summary, carbohydrate metabolism plays a pivotal role in the adaptation of *Lbs. rhamnosus* ATCC 53103 to osmotic stress. Enhanced glucose uptake and utilization through glycolysis and the PTS promote ATP synthesis, thereby sustaining the cellular energy supply. Meanwhile, the regulation of the pyruvate metabolic pathway optimizes the efficiency of energy utilization. Moreover, energy conservation and redox homeostasis are facilitated through the downregulation of glycogen synthesis and genes associated with the pentose phosphate pathway. Collectively, these synergistic regulatory mechanisms reinforce metabolic homeostasis and enhance the stress adaptation capacity of *Lbs. rhamnosus* ATCC 53103.

## 4. Conclusions

In summary, this study, for the first time, combined transcriptomic and proteomic analyses to investigate the response of *Lbs. rhamnosus* ATCC 53103 to osmotic stress induced by 0.6 M sodium lactate. Under these conditions, a total of 792 DEGs and 138 DEPs were identified. The differential regulation of these genes and proteins played a key role in fatty acid metabolism, amino acid metabolism, and carbohydrate metabolism. The research results demonstrate the intrinsic connection between *Lbs. rhamnosus* ATCC 53103 and its metabolite synthesis regulatory network in response to osmotic pressure stress, providing further understanding for the stress response mechanism of lactic acid bacteria exposed to osmotic pressure stress environments, and providing theoretical guidance for addressing the impact of osmotic pressure stress on the production of lactic acid bacteria products. In the future, it is necessary to explore the global response patterns under different stress intensities and combinations to reveal more universal regulatory laws. In addition, combining these molecular mechanisms with industrial fermentation processes is expected to provide new ideas for constructing highly tolerant lactic acid bacteria strains, thereby enhancing the stability and application value of lactic acid bacteria products.

## Figures and Tables

**Figure 1 foods-14-03112-f001:**
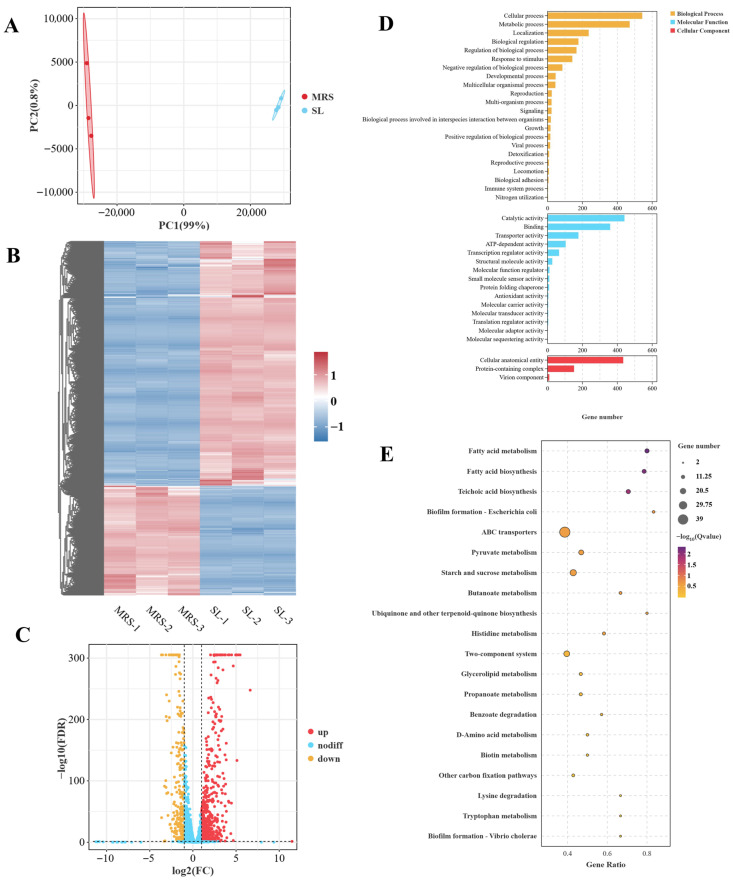
Effect of osmotic stress on the transcriptomics of *Lbs. rhamnosus* ATCC 53103. (**A**) PCA. (**B**) Hierarchical clustering analysis of DEGs in MRS and SL groups (the redder the color, the higher the expression level; the bluer the color, the lower the expression level). (**C**) Volcano plot showing the changes in DEGs before and after osmotic stress in *L. rhamnosus* ATCC 53103. Red dots indicate the upregulation of genes, and yellow dots indicate the downregulation of genes (judgment criteria were FDR < 0.05, fold change ≥ 2). (**D**) GO annotation of DEGs in MRS and SL groups. (**E**) KEGG significance bubble plots of DEGs in MRS and SL groups.

**Figure 2 foods-14-03112-f002:**
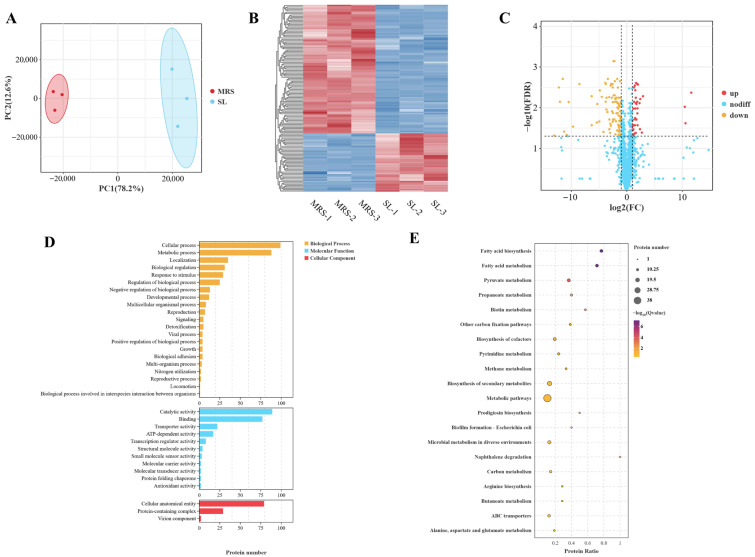
Effect of osmotic stress on the proteomics of *Lbs. rhamnosus* ATCC 53103. (**A**) PCA. (**B**) Hierarchical clustering analysis of DEPs in MRS and SL groups (the redder the color, the higher the expression level; the bluer the color, the lower the expression level). (**C**) Volcano plot showing the changes in DEPs before and after osmotic stress in *Lbs. rhamnosus* ATCC 53103. Red dots indicate the upregulation of proteins, and yellow dots indicate the downregulation of proteins (judged by FDR < 0.05, fold change ≥ 2). (**D**) GO annotation of DEPs in the MRS and SL groups. (**E**) KEGG significance bubble plots of DEPs in MRS and SL groups.

**Figure 3 foods-14-03112-f003:**
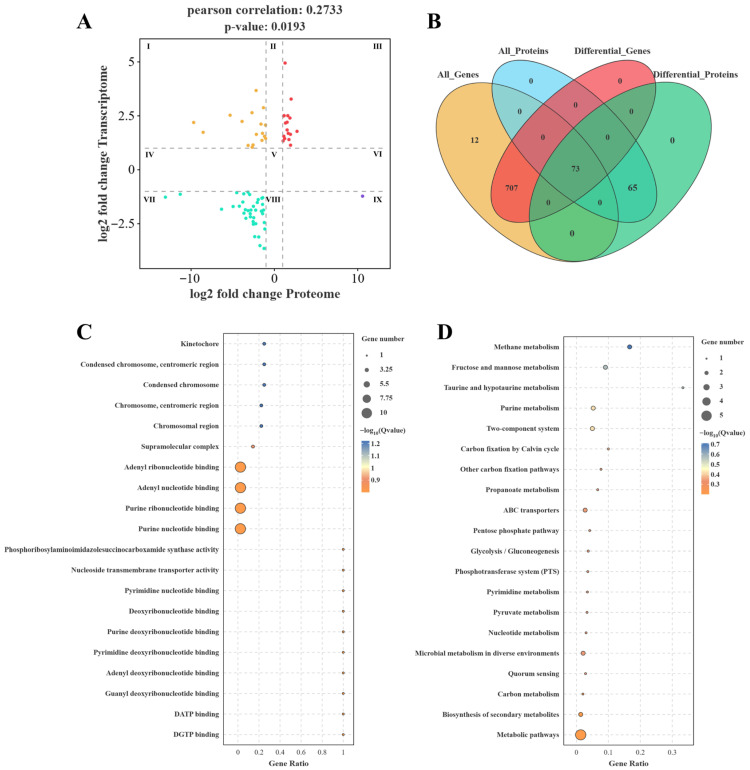
Comprehensive analysis of transcriptomics and proteomics data of *Lbs. rhamnosus* ATCC 53103 cells after osmotic stress (FDR < 0.05, fold change ≥ 2). (**A**) Nine-quadrant plot of transcriptomics versus proteomics, where red dots indicate the consistent upregulation of gene expression in both groups (Genes are mainly distributed in quadrants I, III, VII, and IX. I: The gene is upregulated at the transcriptional level and downregulated at the protein level; III: The gene is upregulated at both the transcriptional and protein levels; VII: The gene is downregulated at both the transcriptional and protein levels; IX: The gene is downregulated at the transcriptional level and upregulated at the protein level). (**B**) Venn diagram showing the distribution of differentially expressed DEGs/DEPs (Each circle represents a set, represented by different colors, and the number of genes contained in each set is marked. The overlapping areas of the circles represent the number of common target genes shared by both sets of analyses, while the non-overlapping areas represent the number of unique target genes specific to each set of analyses). (**C**) GO enrichment of genes and proteins in the third quadrant. (**D**) KEGG enrichment of genes and proteins in the third quadrant.

**Figure 4 foods-14-03112-f004:**
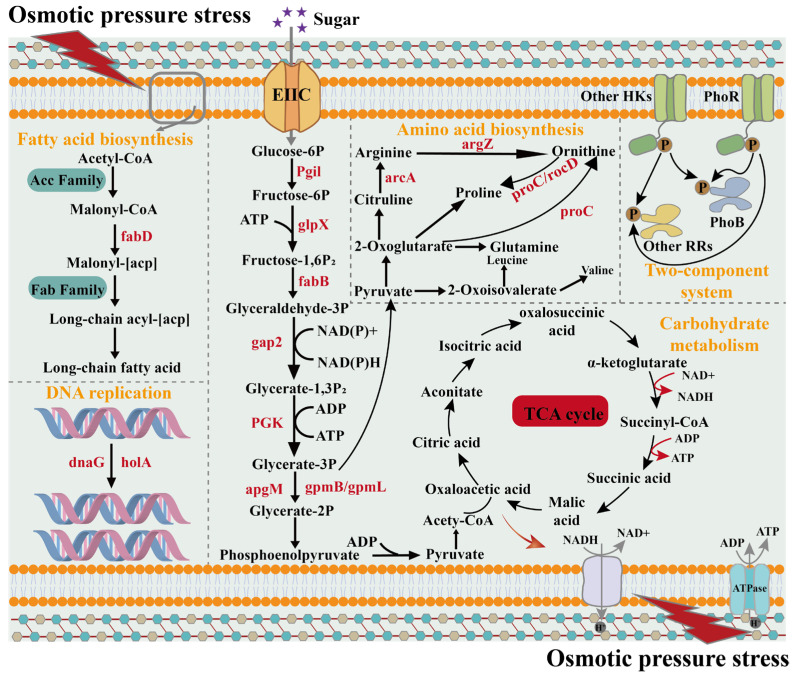
Response mechanism of *Lbs. rhamnosus* ATCC 53103 to osmotic stress.

**Figure 5 foods-14-03112-f005:**
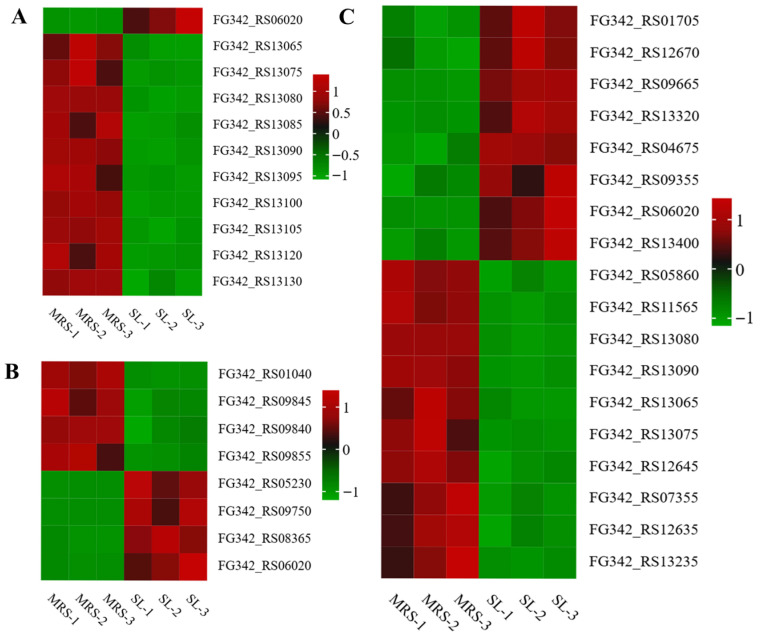
Expression of DEPs associated with osmotic stress tolerance performance. (**A**) Heat map of DEP expression related to fatty acid anabolism. (**B**) Heatmap of DEP expression related to amino acid anabolism. (**C**) Heat map of DEP expression related to carbohydrate metabolism.

**Table 1 foods-14-03112-t001:** DEGs for fatty acid metabolism under osmotic stress.

Gene ID	Symbol	Description	Log2 (FC)
FG342_RS11585	FG342_RS11585	acetyl-CoA C-acyltransferase	1.10
FG342_RS13065	*accA*	ACP S-malonyltransferase	−1.70
FG342_RS13075	FG342_RS13075	carboxyltransferase subunit alpha	−2.18
FG342_RS13080	FG342_RS13080	acetyl-CoA carboxylase biotin carboxyl carrier protein	−1.87
FG342_RS13085	*fabZ*	nitronate monooxygenase	−1.89
FG342_RS13090	*accB*	beta-ketoacyl-ACP synthase II	−1.49
FG342_RS13095	*fabF*	ACP S-malonyltransferase	−2.22
FG342_RS13100	FG342_RS13100	acetyl-CoA carboxylase biotin carboxylase subunit	−2.25
FG342_RS13105	FG342_RS13105	acetyl-CoA carboxylase carboxyltransferase subunit beta	−1.87
FG342_RS13110	FG342_RS13110	nitronate monooxygenase	−1.41
FG342_RS13120	FG342_RS13120	3-hydroxyacyl-ACP dehydratase FabZ family protein	−2.50
FG342_RS13130	FG342_RS13130	beta-ketoacyl-ACP synthase II	−3.10

**Table 2 foods-14-03112-t002:** DEGs for amino acid metabolism under osmotic stress.

Gene ID	Symbol	Description	Log2 (FC)
**Histidine metabolism**			
FG342_RS09740	*hisE*	phosphoribosyl-ATP diphosphatase	−1.46
FG342_RS09745	*hisI*	phosphoribosyl-AMP cyclohydrolase	−1.63
FG342_RS09750	*hisF*	imidazole glycerol phosphate synthase subunit HisF	−1.22
FG342_RS09755	FG342_RS09755	1-(5-phosphoribosyl)-5-[(5- phosphoribosylamino)methylideneamino] imidazole-4-carboxamide isomerase	−1.28
FG342_RS09760	*hisH*	imidazole glycerol phosphate synthase subunit HisH	−1.07
FG342_RS09780	*hisG*	ATP phosphoribosyltransferase	−1.81
FG342_RS09785	FG342_RS09785	ATP phosphoribosyltransferase regulatory subunit	−2.17
**D-Amino acid metabolism**			
FG342_RS03230	FG342_RS03230	D-alanine--D-alanine ligase	3.26
FG342_RS04925	FG342_RS04925	dipeptide epimerase	−1.87
FG342_RS06120	*dltA*	D-alanine--poly(phosphoribitol) ligase subunit DltA	3.09
FG342_RS06130	*dltC*	D-alanine--poly(phosphoribitol) ligase subunit 2	2.96
FG342_RS11445	*purQ*	phosphoribosylformylglycinamidine synthase subunit PurQ	3.61
**Lysine degradation**			
FG342_RS11585	FG342_RS11585	acetyl-CoA C-acyltransferase	1.10
FG342_RS13925	FG342_RS13925	NAD-dependent succinate-semialdehyde dehydrogenase	−1.10
**Tryptophan metabolism**			
FG342_RS11585	FG342_RS11585	acetyl-CoA C-acyltransferase	1.10
FG342_RS13950	FG342_RS13950	amidase	−1.25
**Valine, leucine, and isoleucine degradation**			
FG342_RS03840	FG342_RS03840	NAD(P)-dependent oxidoreductase	−1.41
FG342_RS11575	FG342_RS11575	hydroxymethylglutaryl-CoA synthase	1.01
FG342_RS11585	FG342_RS11585	acetyl-CoA C-acyltransferase	1.10
**Cysteine and methionine metabolism**			
FG342_RS00425	FG342_RS00425	L-lactate dehydrogenase	−1.43
FG342_RS00780	FG342_RS00780	L-2-hydroxyisocaproate dehydrogenase	−1.51
FG342_RS02445	FG342_RS02445	pyridoxal phosphate-dependent aminotransferase	2.28
FG342_RS02865	FG342_RS02865	amidohydrolase family protein	2.29
FG342_RS05985	FG342_RS05985	S-ribosylhomocysteine lyase	−1.63
FG342_RS08580	FG342_RS08580	hypothetical protein	−1.22
FG342_RS08810	FG342_RS08810	GAF domain-containing protein	1.39
FG342_RS09060	FG342_RS09060	5′-methylthioadenosine/adenosylhomocysteine nucleosidase	1.52
FG342_RS13295	FG342_RS13295	homoserine dehydrogenase	−1.01
**Arginine and proline metabolism**			
FG342_RS11720	*proC*	pyrroline-5-carboxylate reductase	1.00
FG342_RS12380	FG342_RS12380	proline-specific peptidase family protein	−1.53
FG342_RS13950	FG342_RS13950	amidase	−1.25
**Arginine biosynthesis**			
FG342_RS04950	FG342_RS04950	glutamine synthetase family protein	−1.87
FG342_RS11445	*purQ*	phosphoribosylformylglycinamidine synthase subunit PurQ	3.61
**Phenylalanine, tyrosine, and tryptophan biosynthesis**			
FG342_RS02965	*trpC*	indole-3-glycerol phosphate synthase TrpC	1.81
FG342_RS02970	*trpD*	anthranilate phosphoribosyltransferase	1.78
**Alanine, aspartate, and glutamate metabolism**			
FG342_RS02450	FG342_RS02450	carbon-nitrogen family hydrolase	2.63
FG342_RS04950	FG342_RS04950	glutamine synthetase family protein	−1.87
FG342_RS11435	*purF*	amidophosphoribosyltransferase	3.48
FG342_RS11445	*purQ*	phosphoribosylformylglycinamidine synthase subunit PurQ	3.61
FG342_RS13925	FG342_RS13925	NAD-dependent succinate-semialdehyde dehydrogenase	−1.10
**Tyrosine metabolism**			
FG342_RS13925	FG342_RS13925	NAD-dependent succinate-semialdehyde dehydrogenase	−1.10
**Phenylalanine metabolism**			
FG342_RS13950	FG342_RS13950	amidase	−1.25
**Lysine biosynthesis**			
FG342_RS01040	FG342_RS01040	PLP-dependent aminotransferase family protein	−2.01
FG342_RS06475	FG342_RS06475	pyridoxal phosphate-dependent aminotransferase	1.12
FG342_RS13295	FG342_RS13295	homoserine dehydrogenase	−1.01
**Glycine, serine, and threonine metabolism**			
FG342_RS13295	FG342_RS13295	homoserine dehydrogenase	−1.01

**Table 3 foods-14-03112-t003:** DEGs for carbohydrate metabolism under osmotic stress.

Gene ID	Symbol	Description	Log2 (FC)
**Glycolysis/Gluconeogenesis**			
FG342_RS00425	FG342_RS00425	L-lactate dehydrogenase	−1.43
FG342_RS00780	FG342_RS00780	L-2-hydroxyisocaproate dehydrogenase	−1.51
FG342_RS01195	FG342_RS01195	glycoside hydrolase family 1 protein	2.22
FG342_RS01430	FG342_RS01430	PTS transporter subunit EIIC	1.25
FG342_RS01450	FG342_RS01450	PTS glucose transporter subunit IIA	2.38
FG342_RS03410	FG342_RS03410	class II fructose-bisphosphate aldolase	−1.71
FG342_RS12670	FG342_RS12670	fructose-bisphosphatase class III	2.51
FG342_RS13435	FG342_RS13435	6-phospho-beta-glucosidase	3.79
**Pyruvate metabolism**			
FG342_RS00425	FG342_RS00425	L-lactate dehydrogenase	−1.43
FG342_RS00780	FG342_RS00780	L-2-hydroxyisocaproate dehydrogenase	−1.51
FG342_RS03270	FG342_RS03270	D-2-hydroxyisocaproate dehydrogenase	1.28
FG342_RS04910	FG342_RS04910	pyruvate oxidase	−2.73
FG342_RS05860	FG342_RS05860	malolactic enzyme	1.45
FG342_RS11565	*spxB*	pyruvate oxidase	−1.32
FG342_RS11585	FG342_RS11585	acetyl-CoA C-acyltransferase	1.10
FG342_RS12000	FG342_RS12000	oxaloacetate decarboxylase subunit alpha	1.34
FG342_RS12035	FG342_RS12035	sodium ion-translocating decarboxylase subunit beta	1.70
FG342_RS13065	*accA*	carboxyltransferase subunit alpha	−1.70
FG342_RS13075	FG342_RS13075	acetyl-CoA carboxylase carboxyltransferase subunit beta	−2.18
FG342_RS13080	FG342_RS13080	acetyl-CoA carboxylase biotin carboxylase subunit	−1.87
FG342_RS13090	*accB*	acetyl-CoA carboxylase biotin carboxyl carrier protein	−1.49
FG342_RS13235	*spxB*	pyruvate oxidase	−2.74
FG342_RS13400	FG342_RS13400	acetate kinase	1.34
**Pentose phosphate pathway**			
FG342_RS01380	*rpiA*	ribose-5-phosphate isomerase RpiA	−1.39
FG342_RS01655	FG342_RS01655	PTS mannose/fructose/sorbose family transporter subunit IID	−1.22
FG342_RS03410	FG342_RS03410	class II fructose-bisphosphate aldolase	−1.71
FG342_RS03465	FG342_RS03465	phosphoketolase family protein	−2.28
FG342_RS03800	*gnd*	phosphogluconate dehydrogenase (NAD(+)-dependent, decarboxylating)	−1.10
FG342_RS03990	*deoC*	deoxyribose-phosphate aldolase	−1.44
FG342_RS03995	FG342_RS03995	phosphopentomutase	−1.10
FG342_RS06695	FG342_RS06695	lactonase family protein	−1.43
FG342_RS12670	FG342_RS12670	fructose-bisphosphatase class III	2.51
**Phosphotransferase system (PTS)**			
FG342_RS01125	FG342_RS01125	PTS mannose/fructose/sorbose transporter subunit IIAB	1.72
FG342_RS01130	FG342_RS01130	PTS sugar transporter subunit IIC	1.43
FG342_RS01135	FG342_RS01135	PTS mannose/fructose/sorbose family transporter subunit IID	1.30
FG342_RS01185	FG342_RS01185	PTS lactose/cellobiose transporter subunit IIA	1.79
FG342_RS01215	FG342_RS01215	PTS transporter subunit EIIC	2.91
FG342_RS01430	FG342_RS01430	PTS transporter subunit EIIC	1.25
FG342_RS01450	FG342_RS01450	PTS glucose transporter subunit IIA	2.38
FG342_RS01520	FG342_RS01520	PTS glucitol/sorbitol transporter subunit IIA	−1.42
FG342_RS01525	FG342_RS01525	PTS glucitol/sorbitol transporter subunit IIB	−1.02
FG342_RS01655	FG342_RS01655	PTS mannose/fructose/sorbose family transporter subunit IID	−1.22
FG342_RS01660	FG342_RS01660	PTS sugar transporter subunit IIC	−1.14
FG342_RS01670	FG342_RS01670	PTS sugar transporter subunit IIA	−1.72
FG342_RS02040	FG342_RS02040	PTS sugar transporter subunit IIB	1.01
FG342_RS02045	FG342_RS02045	PTS lactose/cellobiose transporter subunit IIA	1.21
FG342_RS03115	FG342_RS03115	glucose PTS transporter subunit IIA	3.72
FG342_RS03280	FG342_RS03280	PTS lactose/cellobiose transporter subunit IIA	1.29
FG342_RS04130	FG342_RS04130	PTS transporter subunit EIIC	1.38
FG342_RS05260	FG342_RS05260	PTS glucose transporter subunit IIABC	5.05
FG342_RS05450	FG342_RS05450	lactose-specific PTS transporter subunit EIIC	1.36
FG342_RS05515	FG342_RS05515	PTS transporter subunit IIC	1.62
FG342_RS09350	FG342_RS09350	fructose-specific PTS transporter subunit EIIC	2.13
FG342_RS09355	*pfkB*	1-phosphofructokinase	1.56
FG342_RS12905	FG342_RS12905	PTS sugar transporter subunit IIC	1.11
FG342_RS13440	FG342_RS13440	beta-glucoside-specific PTS transporter subunit IIABC	2.17
**Carbon fixation by Calvin cycle**			
FG342_RS01380	*rpiA*	ribose-5-phosphate isomerase RpiA	−1.39
FG342_RS03410	FG342_RS03410	class II fructose-bisphosphate aldolase	−1.71
FG342_RS12670	FG342_RS12670	fructose-bisphosphatase class III	2.51
**Starch and sucrose metabolism**			
FG342_RS01185	FG342_RS01185	PTS lactose/cellobiose transporter subunit IIA	1.79
FG342_RS01195	FG342_RS01195	glycoside hydrolase family 1 protein	2.22
FG342_RS01215	FG342_RS01215	PTS transporter subunit EIIC	2.91
FG342_RS01300	FG342_RS01300	alpha-glucosidase	−1.14
FG342_RS01425	FG342_RS01425	6-phospho-alpha-glucosidase	1.70
FG342_RS01430	FG342_RS01430	PTS transporter subunit EIIC	1.25
FG342_RS01450	FG342_RS01450	PTS glucose transporter subunit IIA	2.38
FG342_RS02040	FG342_RS02040	PTS sugar transporter subunit IIB	1.01
FG342_RS02045	FG342_RS02045	PTS lactose/cellobiose transporter subunit IIA	1.21
FG342_RS03115	FG342_RS03115	glucose PTS transporter subunit IIA	3.72
FG342_RS03280	FG342_RS03280	PTS lactose/cellobiose transporter subunit IIA	1.29
FG342_RS04130	FG342_RS04130	PTS transporter subunit EIIC	1.38
FG342_RS05255	treC	alpha,alpha-phosphotrehalase	5.29
FG342_RS07070	FG342_RS07070	glycoside hydrolase family 13 protein	−1.79
FG342_RS07075	FG342_RS07075	glycoside hydrolase family 65 protein	−1.80
FG342_RS07080	*pgmB*	beta-phosphoglucomutase	−1.11
FG342_RS12630	FG342_RS12630	glycogen/starch/alpha-glucan phosphorylase	−1.79
FG342_RS12635	*glgA*	glycogen synthase GlgA	−1.91
FG342_RS12640	*glgD*	glucose-1-phosphate adenylyltransferase subunit GlgD	−1.24
FG342_RS12905	FG342_RS12905	PTS sugar transporter subunit IIC	1.11
FG342_RS13435	FG342_RS13435	6-phospho-beta-glucosidase	3.79
**Fructose and mannose metabolism**			
FG342_RS01125	FG342_RS01125	PTS mannose/fructose/sorbose transporter subunit IIAB	1.72
FG342_RS01130	FG342_RS01130	PTS sugar transporter subunit IIC	1.43
FG342_RS01135	FG342_RS01135	PTS mannose/fructose/sorbose family transporter subunit IID	1.30
FG342_RS01270	*rhaB*	rhamnulokinase	1.27
FG342_RS01520	FG342_RS01520	PTS glucitol/sorbitol transporter subunit IIA	−1.42
FG342_RS01525	FG342_RS01525	PTS glucitol/sorbitol transporter subunit IIB	−1.02
FG342_RS01545	FG342_RS01545	SDR family oxidoreductase	−1.16
FG342_RS01660	FG342_RS01660	PTS sugar transporter subunit IIC	−1.14
FG342_RS01670	FG342_RS01670	PTS sugar transporter subunit IIA	−1.72
FG342_RS03410	FG342_RS03410	class II fructose-bisphosphate aldolase	−1.71
FG342_RS09350	FG342_RS09350	fructose-specific PTS transporter subunit EIIC	2.13
FG342_RS09355	*pfkB*	1-phosphofructokinase	1.56
FG342_RS12670	FG342_RS12670	fructose-bisphosphatase class III	2.51
**Galactose metabolism**			
FG342_RS00705	FG342_RS00705	tagatose 1,6-diphosphate aldolase	−1.70
FG342_RS01300	FG342_RS01300	alpha-glucosidase	−1.14
FG342_RS05450	FG342_RS05450	lactose-specific PTS transporter subunit EIIC	1.36
FG342_RS07355	FG342_RS07355	alpha-galactosidase	−3.52

## Data Availability

The original contributions presented in the study are included in the article; further inquiries can be directed to the corresponding authors.

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
