# Peer review of "Integrated Transcriptomic and Proteomic Analyses Revealed the Mechanism of the Osmotic Stress Response in Lacticaseibacillus rhamnosus ATCC 53103"

_foods, 2025, doi:10.3390/foods14173112_

Round 1

Reviewer 1 Report

Comments and Suggestions for Authors

General comments:

In general, I believe that the manuscript does not fall within the scope of Foods. It is a metabolic study of a single strain of a bacterial species. While it is true that this bacterial species is of interest as a prebiotic, it is not studied in any food technology process nor included in any food matrix. Therefore, I believe that the manuscript would be more suitable for the journal Microorganisms.

The work is of little interest from a nutritional point of view. Furthermore, some of the methods are strange, such as the use of DNA in transcriptomics, or the rather meaningless reference to bacteriophages. The presentation is also very sloppy, with many undefined abbreviations in the text and tables without headings on pages 10, 12-13, and 14-18.

Specific comments:

Line 65: Write bacterial names in italics

Please cite the complete name of ATCC in line 82

Line 88: Why bacteriophague are you talking about?

Line 95-96. "The fermentation broth was collected at the end of the loga-95 rithmic growth phase, centrifuged at 8000 rpm at 4 ℃ for 5 min," this phrase was repeated.

Please cite the entor name of "PBS".

Lines 110-113: ADN in transcriptomic analysis?

Line 146: Please define "DIA" the firt time that is cited in the text.

Author Response

Responses to the comments of Reviewer #1

Thank you for your meticulous review of this article and for your valuable suggestions. We would like to further explain the relevance of the manuscript to the "Foods" journal. We sincerely request that you reconsider this matter. This study focuses on the metabolic mechanism of Lacticaseibacillus rhamnosus. This strain has been confirmed by numerous studies to be a core probiotic in yogurt, and its metabolic characteristics directly affect the functional properties and quality of the food. During the production, processing and transportation of fermented foods, Lacticaseibacillus rhamnosus is subject to external environmental stresses such as acid, osmotic pressure and temperature, which leads to a lower bacterial activity. Our work simulates the osmotic stress environment and studies the intrinsic resistance mechanisms within the bacteria through proteomics and transcriptomics. This provides a crucial theoretical basis for optimizing the fermentation food process. It is worth noting that this article covers an important aspect of the characteristics of probiotics, which is applicable to food microbiology. Moreover, this journal has also published similar articles. Although the experimental models are different, these works are consistent with the core objective of this study - to solve key problems in the food industry through basic microbiological research.

  1. Liu, P. F.; Ma, S. Y.; Chen, J.; Duan, C.; Wang, L. X.; Chen, D.; Lv, S. Y.; Li, Y. Z.; Yan, X. L. Fermented sheep milk supplemented with Lactobacillus rhamnosus NM-94: Enhancing fermented milk quality and enriching microbial community in mice. J. Dairy Sci. 2025, 108 (6), 5530-5542.
  2. Zhao, R. T.; Xu, K.; Yan, M. Y.; Peng, J. H.; Liu, H. R.; Huang, S. L.; Zhang, S. S.; Xu, Z. S.; Guo, X. P.; Wang, T. Preparation of sweet milk and yogurt containing d-tagatose by the l-arabinose isomerase derived from Lactobacillus rhamnosus. Lwt-Food Science and Technology 2023, 187, 115355.

Thank you very much for these perfect and specific suggestions you provided regarding improving the readability of our manuscript. We have made detailed revisions to the entire text and also modified the grammar. The content provided and revised in the manuscript has been marked in red font.

Comments 1.

The work is of little interest from a nutritional point of view. Furthermore, some of the methods are strange, such as the use of DNA in transcriptomics, or the rather meaningless reference to bacteriophages. The presentation is also very sloppy, with many undefined abbreviations in the text and tables without headings on pages 10, 12-13, and 14-18.

Response: Sorry, it was our mistake. All the tables have been given comprehensive titles, as follows:

Table 1 (Page 11): DEGs for fatty acid metabolism under osmotic stress.

Table 2 (Pages 12-14): DEGs for amino acid metabolism under osmotic stress.

Table 3 (Pages 15-19): DEGs for carbohydrate metabolism under osmotic stress.

During the revision process, we have systematically standardized all professional terms to ensure that each term is presented in its full form for the first time, and subsequent references use the standard abbreviated form.

Comments 2.

Line 65: Write bacterial names in italics.

Response: Thank you for raising this point. We have revised and replaced the case study and uniformly proofread all microbiological nomenclature throughout the text to ensure that all bacterial strain names adhere to the standardized italicized format.

Comments 3.

Please cite the complete name of ATCC in line 82.

Response: We thank you for this comment. After carefully reviewing the content of line 82 and conducting a systematic check of the entire text, we did not find any expressions related to strain naming in the designated location. However, we have thoroughly rechecked all the microbial-related naming in the entire text and have made corresponding revisions to ensure the accuracy and completeness of the naming. Mainly includes: Line 83, Line 96, Line 100, Line 101, Line 102, Line 201, Line 203, Line 207, Line 213, Line 217, Line 236, Line 243, Line 249, Line 255, Line 265, Line 269, Line 271, Line 276, Line 295, Line 300, Line 304, Line 307, Line 314, Line 315, Line 344, Line 356, Line 394, Line 401, Line 452, Line 459, Line 468, Line 472.

Comments 4.

Line 88: Why bacteriophague are you talking about?

Response: Sorry, it was our mistake. After verification, it was found that due to a typing error, "bacteria" was mistakenly written as "bacteriophague" in the text. This is indeed an oversight in our expression. In the revised version, we have corrected all relevant expressions to "bacteria" and conducted a systematic review of the entire text to ensure the accuracy of the terminology usage. Lines 106-107: After 24 h, the bacteria cells were harvested for further analysis.

Comments 5.

Line 95-96. "The fermentation broth was collected at the end of the loga-95 rithmic growth phase, centrifuged at 8000 × g at 4 ℃ for 5 min," this phrase was repeated.

Response: We have revised it as suggested. We have changed "centrifuged at 8000 × g at 4 ℃ for 5 min" to "centrifugal (as mentioned above)". Lines 112-114: The fermentation broth was collected at the end of the logarithmic growth phase, centrifugal (as mentioned above), and the pellet was collected.

Comments 6.

Please cite the entor name of "PBS".

Response: Sorry, it was our mistake. We have changed "PBS" to "phosphate buffered saline (PBS)". Lines 114-116: The pellet was washed three times with phosphate buffered saline (PBS) (Yuanye, Shanghai yuanye Bio-Technology Co., Ltd), frozen in liquid nitrogen, and stored at -80 ℃ until further analysis.

Comments 7.

Lines 110-113: ADN in transcriptomic analysis?

Response: Thank you for raising this point. The DNA in lines 132-145 is deoxyribonucleic acid (DNA), one of the four macromolecules found in biological cells and a type of nucleic acid. DNA is a nucleic acid that carries the genetic information necessary for the synthesis of RNA and proteins, and is an essential macromolecule for the development and normal functioning of living organisms.

In transcriptomics, the synthesis of complementary DNA (cDNA) is a crucial step in RNA sequencing (RNA-seq) and other analyses. RNA is prone to degradation by RNAase and becomes unstable at high temperatures (such as during PCR amplification). By reverse transcription, RNA is converted into a more stable DNA form, which is convenient for long-term storage and subsequent operations. Moreover, the second-generation sequencers (Illumina) are specifically designed for DNA, and cDNA can be directly used for library construction.

The specific process of library construction is as follows: Total RNA→Remove rRNA→RNA fragment (200 nt)→Random hexamer primer cDNA synthesis→USER Enzyme→Size selection and PCR amplification→Illumina sequencing.

  1. Zhao, J. N.; Kwok, L. Y.; Fan, H.; Liu, X. F.; Yongfu Chen, Y. F. Analyses of cellular responses of phorate-incubated Lactiplantibacillus plantarum by transcriptomics and proteomics. LWT. 2025, 189, 115443.
  2. Huan Yang, H.; Zhang, L.; Li, J. S.; Jin, Y.; Zou, J. P.; Huang, J.; Zhou, R. Q.; Huang, M. Q.; Wu, C. D. Cell surface properties and transcriptomic analysis of cross protection provided between heat adaptation and acid stress in Tetragenococcus halophilus. Food Research International. 2021, 140, 110005.
  3. Liu, M. M.; Feng, M. X.; Yang, K.; Cao, Y. F.; Zhang, J.; Xu, J. N.; Hernández, S. H.; Wei, X. Y.; Fan, M. T. Transcriptomic and metabolomic analyses reveal antibacterial mechanism of astringent persimmon tannin against Methicillin-resistant Staphylococcus aureus isolated from pork. Food Chemistry. 2020, 309, 125692.

Comments 8.

Line 146: Please define "DIA" the firt time that is cited in the text.

Response: Sorry, it was our mistake. We have changed "DIA" to "data independent acquisition (DIA)". Lines 183-185: The mass spectrometer was operated in diaPASEF mode to acquire data independent acquisition (DIA) data, with a scanning range of 349-1229 m/z and an isolation window width of 40 Da.

Reviewer 2 Report

Comments and Suggestions for Authors

This manuscript investigates the transcriptomic and proteomic responses of Lacticaseibacillus rhamnosus ATCC 53103 to sodium lactate–induced osmotic stress. The integration of multi-omics approaches provides potentially valuable insights into stress adaptation in lactic acid bacteria. However, several sections would benefit from clarification, stronger justification, and refinement to avoid overgeneralization or unsupported claims. Below are my detailed comments organized by manuscript sections.

Abstract

Line 15–24: The abstract clearly summarizes findings, but some claims (e.g., “enhanced osmotic stress resistance observed”) are too strong given that actual resistance/survival data were not measured. Please rephrase to emphasize observed transcriptional and proteomic changes rather than functional outcomes.

Introduction
• Line 31–33: The statement that LAB have applications in “cell factory construction and disease treatment” is overgeneralized. Consider providing references or limiting the scope to well-established uses.
• Line 37–39: The claim that L. rhamnosus “improves hyperglycemia and insulin resistance” is overstated without context (strain-specific or clinical evidence). Please qualify or soften this.
• Line 42–44: Assertions about immune regulation and allergy mitigation are broad and read as health claims. Please provide specific supporting references or narrow the scope.
• Line 45–55: The transition from acid stress to osmotic stress caused by sodium lactate is abrupt. Stronger justification is needed for why osmotic stress (via sodium lactate accumulation) is the focal stressor.
• Line 56–68: Examples from Corynebacterium and Bacillus species are informative, but the relevance to LAB is not clearly explained. Please clarify why these analogies are appropriate.
• Line 77–78: “Practical application of LAB fermentation products” is vague. Consider specifying whether this refers to improved industrial fermentation stability, probiotic survival, or other outcomes.

Materials and Methods
• Line 87–88: The text refers to “bacteriophage cells,” which appears to be a misstatement. This should be corrected to “bacterial cells.”
• Line 95: The sentence “the fermentation broth was collected at the end of the logarithmic growth phase” is repeated redundantly. Please edit for clarity.
• Line 86: The choice of 0.6 M sodium lactate as the osmotic stress condition is not justified. Please explain why this concentration was selected and whether it is physiologically relevant.
• Line 87: The use of a single 24 h timepoint limits interpretation. If this was a design choice, please justify why dynamic responses were not investigated.
• Line 99: Clarify whether the three replicates represent biological or technical replicates.
• Lines 102–113 vs. 126–159: Transcriptomics methods are relatively concise, whereas proteomics methods are described in excessive operational detail. For balance and readability, consider streamlining instrument/gradient specifics to supplementary information.
• Line 167: Local normalization strategy is mentioned but not explained. Please provide a brief rationale for this choice.

Results and Discussion 
• Line 205–207: Editorial placeholder text remains (“This section may be divided by subheadings…”). This should be removed.
• Line 225–226: The statement about “pronounced alterations” repeats the obvious from PCA and clustering without deeper analysis. Consider condensing.
• Line 261–263: A Pearson correlation of 0.27 is weak, not moderate. Please revise the interpretation.
• Line 269–270: “Enhance adaptability” is an overstatement. The data show enrichment in energy-related pathways, but functional adaptation was not directly tested.
• Line 294–296: The claim that fatty acid degradation rates are reduced is speculative, as no flux or metabolite measurements were performed.
• Line 301–302: Reference to effects on “organic acid concentrations” is unsupported, since organic acids were not measured.
• Line 306–308: The suggestion that acetyl-CoA binding to AdhE channels it into fatty acid biosynthesis is speculative. Please revise.
• Line 326–327 vs. 328–329: The text first suggests histidine contributes to gluconeogenesis but then reports histidine biosynthesis downregulation. This is inconsistent and requires clarification.
• Line 339–340: Compensation through acetyl-CoA channeling is speculative. Please soften this claim.
• Line 344–345: Intracellular pH regulation is hypothesized, but no pH data were collected. Please acknowledge this as speculation.
• Line 360–363: Citation on glutamine metabolism refers to Lactobacillus broadly, but your results show downregulation in L. rhamnosus. Please reconcile this discrepancy.
• Line 366–372: The claim that integrated analyses “conferred a robust capacity” is overstated, since survival or resistance outcomes were not measured.
• Line 388–390 vs. 425–427: The text states glycogen resynthesis may occur but later acknowledges glycogen synthase downregulation. Please resolve this inconsistency.
• Line 404–407: Discussion of acetyl phosphate acetylating DNA/proteins is irrelevant to your data. Suggest removing.
• Line 411–412: Statements on pyruvate flow to other branches are speculative without metabolic flux analysis.
• Line 419–420: NADPH levels were not measured, so claims of diminished reducing power should be qualified.
• Line 427–428: “Enhanced resistance to osmotic stress” is overstated; no viability data are presented.
• Line 429–432: Conclusions about “metabolic flexibility” and “sustained energy supply” are broad. Please reframe to emphasize observed gene/protein expression trends rather than functional outcomes.

Conclusion

The conclusion largely repeats the abstract. Consider reframing to highlight the novelty of your multi-omics findings and their limitations. For example, emphasize the observed transcriptional/proteomic shifts, while noting that physiological validation (e.g., growth, survival, metabolite assays) remains to be established.

Author Response

Responses to the comments of Reviewer #2

We sincerely appreciate your excellent and detailed suggestions for improving our manuscript's readability. The supplied content and manuscript revisions have been highlighted in red text.

Comments 1.

Abstract:

Line 15-24: The abstract clearly summarizes findings, but some claims (e.g., "enhanced osmotic stress resistance observed") are too strong given that actual resistance/survival data were not measured. Please rephrase to emphasize observed transcriptional and proteomic changes rather than functional outcomes.

Response: Thank you for catching this. We have made modifications to the absolute content expressed in this part. Lines 19-27: Differential regulation of these genes/proteins mainly includes inhibition of fatty acid metabolism with membrane structural remodeling (down-regulation of ACC family and FAB family expression), dynamic homeostasis of amino acid metabolism (restriction of synthesis of histidine, cysteine, leucine, etc., and enhancement of catabolism of lysine, tryptophan, etc.), and survival oriented reconfiguration of carbohydrate metabolism (gene expression related to the glycolytic pathway increases, while gene expression related to the pentose phosphate pathway decreases). These synergistic alterations in metabolic regulation may facilitate the adaptive response of Lbs. rhamnosus ATCC 53103 to osmotic stress.

Comments 2.

Introduction

(1) Line 31-33: The statement that LAB have applications in "cell factory construction and disease treatment" is overgeneralized. Consider providing references or limiting the scope to well-established uses.

Response: Based on your suggestion, we have made the necessary revisions to make it more accurate. Lines 36-37: Due to their unique functional characteristics, LAB have significant applications in food and medicine.

(2) Line 37-39: The claim that L. rhamnosus "improves hyperglycemia and insulin resistance" is overstated without context (strain-specific or clinical evidence). Please qualify or soften this.

Response: We thank you for this comment. We agree that the expression "improves hyperglycemia and insulin resistance" in the original text is indeed too absolute. We have modified it to be more cautious. Lines 41-43: According to existing research, certain strains of Lbs. rhamnosus may alleviate specific types of diarrhea, such as antibiotic-associated diarrhea [3]. Some studies also suggest that these LAB may have positive effects on glucose metabolism and insulin sensitivity [4].

(3) Line 42-44: Assertions about immune regulation and allergy mitigation are broad and read as health claims. Please provide specific supporting references or narrow the scope.

Response: Thank you for your important correction on the accuracy of your expression about immune regulation. We have made the following modifications to the relevant expressions. Lines 46-48: Additionally, certain strains of Lbs. rhamnosus may play a role in immune regulation and have potential associations with allergic reaction regulation.

(4) Line 45-55: The transition from acid stress to osmotic stress caused by sodium lactate is abrupt. Stronger justification is needed for why osmotic stress (via sodium lactate accumulation) is the focal stressor.

Response: Thank you for raising this point. We will answer the questions you raised. Because lactic acid bacteria produce organic acids during the fermentation process, this intracellular acidification has adverse effects on the physiological functions of the cells. To maintain a stable environment during the fermentation process, exogenous neutralizers (such as sodium hydroxide) are usually introduced to counteract the accumulation of lactic acid and reduce its inhibitory effect on the lactic acid bacteria. However, excessive neutralization can lead to the accumulation of sodium lactate, resulting in an increase in the osmotic pressure in the environment, which in turn affects the bacteria. The core objective of this study is to elucidate the impact of osmotic pressure stress on the metabolic status of lactic acid bacteria, rather than comparing its significance with other types of stress. To achieve this, we added sodium lactate exogenously to the culture medium to simulate the high osmotic pressure conditions that lactic acid bacteria might encounter in the actual fermentation environment. In this experimental system, we focused on investigating the regulatory effects of osmotic pressure stress on the gene and protein expression levels of lactic acid bacteria, thereby revealing its potential adaptation mechanisms and energy metabolism changes.

(5) Line 56-68: Examples from Corynebacterium and Bacillus species are informative, but the relevance to LAB is not clearly explained. Please clarify why these analogies are appropriate.

Response: Thank you for catching this. By systematically reviewing the literature cited in the introduction, we found that literature [9-11] failed to fully explain the molecular response mechanism of lactic acid bacteria under osmotic stress conditions, especially lacking in-depth data support and mechanism analysis at the levels of gene expression regulation and protein regulation. For this reason, we conducted a large-scale literature search and strict screening again, replacing and updating these three key references. The newly added literature, based on transcriptomics or proteomics research methods, has comprehensively revealed the molecular regulatory pathways of lactic acid bacteria in response to osmotic stress, including changes in key gene expression and functional regulation of related proteins, thereby providing more solid and comprehensive theoretical support for this study.

Lines 60-79: For example, under osmotic stress, the genes GshAB, GshR3, PepN, GshR4, and serA, which are associated with amino acid metabolism, are upregulated in Lactobacillus plantarum FS5-5. In addition, I526_2330, Gpd, and Gnd, which participate in carbohydrate metabolism, also show increased expression. Lactobacillus plantarum FS5-5 protects cellular proteins and macromolecules by accumulating compatible solutes, elevating GSH levels, and upregulating DNA repair proteins [9]. Another study found that under osmotic pressure stress, Lacticaseibacillus paracasei Zhang transformed into a viable but non-culturable (VBNC) state. Moreover, while maintaining cell integrity, the loss of surface smoothness and the increase in cell aggregation were observed. Further transcriptomic analysis revealed that genes related to carbohydrate and nutrient transport (such as PTS and ABC transport proteins) were upregulated, suggesting that this strain may adapt to osmotic pressure stress and maintain survival by enhancing substance uptake and metabolic capabilities [10]. Additionally, nontargeted metabolomic analysis of Bifidobacterium bifidum CCFM16 during osmotic adaptation revealed an upregulation of F6PPK, a key enzyme in the bifid shunt pathway. This finding suggests that cells redirected energy from basal metabolism toward the synthesis of osmoprotectants to mitigate osmotic stress. Under prolonged hyperosmotic stress, Bifidobacterium bifidum CCFM16 developed a protective mechanism by converting glutamic acid to proline, thereby establishing an osmoprotective system that primarily relies on proline as the adaptive solute [11].

  1. Li, M.; Wang, Q. Q.; Song, X. F.; Guo, J. J.; Wu, J. R.; Wu, R. N. iTRAQ-based proteomic analysis of responses of Lactobacillus plantarum FS5-5 to salt tolerance. Ann. Microbiol. 2019, 69, 377-394.
  2. Bao, Q. H.; Yuan, B. Y.; Ma, X. B.; Zhao, X.; Gao, R.; Jianan Li, J. N.; Kwok, L. Y. Osmotic and cold stress-induced viable but non-culturable state in Lacticaseibacillus paracasei Zhang: A transcriptome analysis. Int. Dairy J. 2025, 166, 106228.
  3. Ying Zhang, Y.; Mao, B. Y.; Tang, X.; Liu, X. M.; Zhao, J. X.; Zhang, H.; Cui, S. M.; Chen, W. Integrative genome and metabolome analysis reveal the potential mechanism of osmotic stress tolerance in Bifidobacterium bifidum. Lwt-Food Science and Technology 2022, 159, 113199.

(6) Line 77-78: "Practical application of LAB fermentation products" is vague. Consider specifying whether this refers to improved industrial fermentation stability, probiotic survival, or other outcomes.

Response: We thank you for this comment. Lines 87-93: By elucidating the molecular response mechanisms of lactic acid bacteria under osmotic stress, this study provides a theoretical foundation for optimizing production processes in the food industry, including fermented dairy products and pickled vegetables, as well as for developing high-value-added products such as extracellular polysaccharides and probiotic preparations. These findings also offer practical guidance for addressing environmental stress fluctuations and improving the tolerance of bacterial strains.

Comments 3.

Materials and Methods

(1) Line 87-88: The text refers to "bacteriophage cells", which appears to be a misstatement. This should be corrected to "bacterial cells".

Response: Sorry, it was our mistake. After verification, it was found that due to a typing error, "bacteria" was mistakenly written as "bacteriophague" in the text. This is indeed an oversight in our expression. In the revised version, we have corrected all relevant expressions to "bacteria" and conducted a systematic review of the entire text to ensure the accuracy of the terminology usage. Lines 106-107: After 24 h, the bacteria cells were harvested for further analysis.

(2) Line 95: The sentence "the fermentation broth was collected at the end of the logarithmic growth phase" is repeated redundantly. Please edit for clarity.

Response: Thank you for raising this point. We have changed "the fermentation broth was collected at the end of the logarithmic growth phase" to "collect cells during the logarithmic growth phase". Lines 109-111: For the transcriptomics analysis: collect cells during the logarithmic growth phase, then centrifuged at 8000 × g for 5 min at 4 ℃. The resulting pellet was frozen in liquid nitrogen and stored at -80 ℃ until further analysis.

(3) Line 86: The choice of 0.6 M sodium lactate as the osmotic stress condition is not justified. Please explain why this concentration was selected and whether it is physiologically relevant.

Response: We thank you for this comment. Based on the systematic screening in the pre-experiment, a gradient of sodium lactate concentration (0, 0.2, 0.4, 0.6, 0.8, 1.0 M) was established. The cell survival status was evaluated by monitoring the OD600 value (biomass) and the plate viable count (CFU/mL). The sub-lethal concentration of sodium lactate was determined to be 0.6 M. Therefore, in the subsequent experiments, 0.6 M concentration of sodium lactate was used as the osmotic stress.

  1. Luo, W.; Zhuang, Y. L.; Sun, L. P.; Gu, Y.; Ding, Y. Y.; Fan, X. J. Regulation of proline on Lacticaseibacillus rhamnosus cells under sodium lactate mediated osmotic stress: resistance and underlying mechanisms. Food Research International. 2025, 221 (2), 117356. Doi: 10.1016/j.foodres. 2025.117356.

(4) Line 87: The use of a single 24 h timepoint limits interpretation. If this was a design choice, please justify why dynamic responses were not investigated.

Response: Thank you for catching this. Based on the previous experiments, the growth of the bacteria under osmotic stress mediated by sodium lactate was analyzed (with OD600 measured every 2 h). It was found that the logarithmic growth phase of the bacteria ended at 24 h. Therefore, the bacteria cultured for 24 h were selected for subsequent analysis.

  1. Luo, W.; Zhuang, Y. L.; Sun, L. P.; Gu, Y.; Ding, Y. Y.; Fan, X. J. Regulation of proline on Lacticaseibacillus rhamnosus cells under sodium lactate mediated osmotic stress: resistance and underlying mechanisms. Food Research International. 2025, 221 (2), 117356. Doi: 10.1016/j.foodres.2025.117356.

(5) Line 99: Clarify whether the three replicates represent biological or technical replicates.

Response: Sorry, it was our mistake. We have made revisions to this passage. Lines 117-118: Three biological replicates were set up for each group. All the samples were frozen and stored for subsequent detection and analysis.

(6) Lines 102-113 vs. 126-159: Transcriptomics methods are relatively concise, whereas proteomics methods are described in excessive operational detail. For balance and readability, consider streamlining instrument/gradient specifics to supplementary information.

Response: Thank you for catching this. By comparing and analyzing the methodological descriptions of transcriptomics and proteomics, we found that there is a problem of insufficient details in the transcriptomics part. In the revised draft, the experimental procedures of this part have been systematically supplemented and improved. Lines 119-153:

2.3 Transcriptomics analysis by RNA sequencing (RNA-seq)

2.3.1 RNA extraction and preprocessing

The sample was ground in liquid nitrogen using an appropriate amount. Trizol reagent (Life Technologies, USA) was added, mixed thoroughly, and incubated for 10 min to ensure complete cell lysis. Chloroform (Guangzhou Chemical Reagent Factory) was added, mixed by inversion, and centrifuged at 14,000 × g for 10 min at 4  ℃. The mixture separated into organic and aqueous phases. The upper aqueous phase was collected, and equal volumes of chloroform and isopropanol (Guangzhou Chemical Reagent Factory) were added. Following centrifugation, the supernatant was discarded, and the pellet was washed with 75 % ethanol (Guangzhou Chemical Reagent Factory), vacuum-dried for 2 - 4 min, and resuspended in an appropriate volume of RNase free water. The solution was incubated at room temperature for 10 min to ensure complete dissolution, then mixed thoroughly, briefly centrifuged, and stored at -80 ℃.

2.3.2 Library construction, quality control, and sequencing

Total RNA was purified using Agencourt RNA Clean XP Beads (Agencourt Bioscience, USA) to deplete ribosomal RNA (rRNA). First- and second-strand cDNA syn-thesis was performed using a PCR thermal cycler. The resulting double-stranded cDNA was purified using 1.8 × Agencourt AMPure XP Beads (Agencourt Bioscience, USA), and the supernatant was collected. Adapter ligation and end repair were sub-sequently performed using a PCR thermal cycler (Dongsheng Xingye Scientific Instruments Co., Ltd., ETC811). USER enzyme was added, and the mixture was incubated at 37  ℃ for 15 min. The product was then purified using AMPure XP Beads (Agencourt Bioscience, USA), washed with 80 % ethanol, eluted with ddHâ‚‚O, and amplified by PCR, followed by a final purification. Library quality was assessed using the ABI StepOnePlus Real-Time PCR System (Life Technologies, USA). The final library was sequenced on the Illumina NovaSeq X Plus platform.

2.3.3 Analysis of differentially expressed genes in RNA-seq datasets

To compare gene expression levels across genes and samples, the dataset was normalized using the fragments per kilobase of transcript per million mapped reads (FPKM) method [14]. Gene expression levels were estimated using RSEM software and normalized using the FPKM method to eliminate the effects of gene length and sequencing depth. Differentially expressed genes (DEGs) were identified using the edgeR package (version 3.12.1), with a fold change ≥ 2 and a false discovery rate (FDR) < 0.05 as the screening threshold. Functional enrichment analysis of DEGs was performed using Gene Ontology (GO) terms and Kyoto Encyclopedia of Genes and Genomes (KEGG) pathways, with a significance threshold of q-value < 0.05.

(7) Line 167: Local normalization strategy is mentioned but not explained. Please provide a brief rationale for this choice.

Response: Sorry, it was our mistake. After careful review, we have confirmed that there are issues with the clarity of the description in Section 2.4.3 of the original manuscript. To enhance the accuracy and completeness of the description, we have made modifications to this part. Lines 189-200: The Data of DIA were processed and analyzed by Spectronaut 18 (Biognosys AG, Switzerland) with default settings. Specific trypsin was set as the digestion type and digestion enzyme. Carbamidomethyl on cysteine was specified as the fixed modification. Oxidation on methionine was specified as the variable modifications. Retention time prediction type was set to dynamic iRT. Data extraction wasdetermined by Spectronaut based on the extensive mass calibration. Spectronaut will determine the ideal extraction window dynamically depending on iRT calibration and gradient stability. Qvalue (FDR) cutoff on precursor level was 1 % and protein level was 1 %. Decoy generation was set to mutated which similar to scrambled but will only apply a random number of AA position swamps (min = 2, max = length/2). Normalization strategy was set to Local normalization. Peptides which passed the 1 % Qvalue cut off were used to calculate the major group quantities with MaxLFQ method.

Comments 4.

Results and Discussion

(1) Line 205-207: Editorial placeholder text remains ("This section may be divided by subheadings…"). This should be removed.

Response: Sorry, it was our mistake. Based on your valuable suggestions, the designated content in the text has been deleted.

(2) Line 225-226: The statement about "pronounced alterations" repeats the obvious from PCA and clustering without deeper analysis. Consider condensing.

Response: Thank you for bringing this issue to our attention. We have made the following revisions to the original text: The sentence "This finding suggests that Lbs. rhamnosus ATCC 53103 undergoes pronounced alterations in protein expression patterns in response to osmotic stress" has been removed.

(3) Line 261-263: A Pearson correlation of 0.27 is weak, not moderate. Please revise the interpretation.

Response: Sorry, it was our mistake. Lines 286-290: As shown in Figure 3A, there is a significant but weak positive correlation between mRNA and protein expression levels (Pearson's r = 0.273, p < 0.05), indicating that under stress conditions, there is limited coordination between the transcriptional and proteomic responses.

(4) Line 269-270: "Enhance adaptability" is an overstatement. The data show enrichment in energy-related pathways, but functional adaptation was not directly tested.

Response: Thank you for pointing out the issue of this statement not being rigorous enough. We have revised this statement. Lines 296-298: These research results suggest that Lbs. rhamnosus ATCC 53103 may improve its adaptability to hypertonic environments to a certain extent by regulating energy metabolic pathways (Figure 3C).

(5) Line 294-296: The claim that fatty acid degradation rates are reduced is speculative, as no flux or metabolite measurements were performed.

Response: We thank you for this comment. The statement lacks direct experimental evidence and is overly speculative. Lines 320-322: This finding indicates that in a high osmotic pressure environment, cells may inhibit the synthesis of fatty acids to save energy and metabolic resources, which is part of an adaptive regulatory strategy.

(6) Line 301-302: Reference to effects on "organic acid concentrations" is unsupported, since organic acids were not measured.

Response: According to your advices, we have removed the original statement about the concentration of organic acids. We are grateful for this suggestion, which helps to enhance the rigor of the manuscript.

(7) Line 306-308: The suggestion that acetyl-CoA binding to AdhE channels it into fatty acid biosynthesis is speculative. Please revise.

Response: According to your suggestion, we have revised the overly absolute expressions in the text. Lines 330-334: Proteomics analysis (Figure 5A and Table S4) revealed that the expression of AdhE, which encodes the bifunctional acetaldehyde-CoA/alcohol dehydrogenase, was upregulated. This might reflect the adaptive response of the cells to the damage induced by osmotic stress or the imbalance of energy metabolism.

(8) Line 326-327 vs. 328-329: The text first suggests histidine contributes to gluconeogenesis but then reports histidine biosynthesis downregulation. This is inconsistent and requires clarification.

Response: Sorry, it was our mistake. Due to the deviation in literature comprehension and the omissions during the writing process, the original expression did not match the intended expression. We re-examined the relevant references and systematically reorganized and corrected this part. Lines 351-355: Histidine is a glycogenic amino acid that enters the gluconeogenic pathway under adverse conditions. All seven differentially expressed genes involved in the histidine metabolic pathway were downregulated under osmotic stress, suggesting that histidine may reduce endogenous glucose production by suppressing its own synthesis.

(9) Line 339-340: Compensation through acetyl-CoA channeling is speculative. Please soften this claim.

Response: We thank you for this comment. Lines 366-367: This metabolic regulatory function may help offset the energy and material deficiencies resulting from reduced fatty acid synthesis capacity.

(10) Line 344-345: Intracellular pH regulation is hypothesized, but no pH data were collected. Please acknowledge this as speculation.

Response: According to your suggestions, we have made the necessary revisions to the relevant statements. Lines 371-373: This suggest that the enzyme may regulate the intracellular environment by reducing acidification within cells in the experimental group and modulate energy production to promote cell growth.

(11) Line 360-363: Citation on glutamine metabolism refers to Lactobacillus broadly, but your results show downregulation in L. rhamnosus. Please reconcile this discrepancy.

Response: Sorry, it was our mistake. After a thorough review of this case, we found that the connection between it and the discussion in this part is relatively insufficient. In view of this, we have updated the case and made adjustments to the corresponding content to enhance the rigor of the argument and the consistency of the overall logic. Lines 388-394: Within the proteomic profile associated with amino acid transport and metabolism, the expression of carbamoyl phosphate synthase (CarB) exhibited a pronounced downregulation. This enzyme plays a pivotal role in the biosynthesis of L-arginine via the ornithine cycle and also serves as a critical component in pyrimidine nucleotide synthesis. Consequently, the downregulation of CarB expression may constrain intracellular arginine biosynthesis and perturb pyrimidine metabolism, thereby influencing associated cellular processes.

(12) Line 366-372: The claim that integrated analyses "conferred a robust capacity" is overstated, since survival or resistance outcomes were not measured.

Response: We are grateful to the reviewer for pointing out this issue. We agree that the original statement was too absolute and have revised the relevant descriptions accordingly. Lines 398-400: This metabolic reprogramming may temporarily suppress certain amino acid biosynthetic pathways. However, it may also play a positive role in maintaining essential cellular functions and adapting to hyperosmotic stress.

(13) Line 388-390 vs. 425-427: The text states glycogen resynthesis may occur but later acknowledges glycogen synthase downregulation. Please resolve this inconsistency.

Response: Sorry, it was our mistake. After re-reviewing the relevant content and conducting a literature check, we discovered that due to misunderstandings in the interpretation of some data during the writing process, there were inconsistencies in two statements in the text. After discussion and referring to relevant literature evidence, we have now deleted the statements lacking experimental data support [specific content: Transcriptomic and proteomic analyses revealed that the expression of the gene and protein encoding fructose-bisphosphatase class III (FG342_RS12670) was upregulated. This suggests that, following osmotic stress, Lbs. rhamnosus ATCC 53103 may enhance the reverse synthesis of glucose-6-phosphate from the glycolytic intermediate fructose-1,6-bisphosphate, enabling the re-synthesis of glycogen or its redistribution to other biosynthetic pathways.] to ensure the rigor and accuracy of the research conclusion.

(14) Line 404-407: Discussion of acetyl phosphate acetylating DNA/proteins is irrelevant to your data. Suggest removing.

Response: We have revised it as suggested. Thanks. We have removed this part of the content from the revised version.

(15) Line 411-412: Statements on pyruvate flow to other branches are speculative without metabolic flux analysis.

Response: Sorry, it was our mistake. We have removed this part.

(16) Line 419-420: NADPH levels were not measured, so claims of diminished reducing power should be qualified.

Response: Sorry, it was our mistake. Given the lack of direct measurement data on NADPH content in the current research, and considering the problem of over-interpretation in the original text, we have revised the relevant statements to make them more rigorous and accurate. Lines 441-444: Specifically, gnd in the oxidative phase, as well as rpiA and FG342_RS03995 (encoding phosphopentomutase) in the non-oxidative phase, were downregulated. This suppression may reduce ribulose-5-phosphate synthesis, leading to a disruption in redox balance.

(17) Line 427-428: "Enhanced resistance to osmotic stress" is overstated; no viability data are presented.

Response: According to your suggestion, we have revised the overly absolute expressions in the text. Lines 450-452: This inhibitory effect may promote energy conservation in bacteria by suppressing glycogen synthesis and maintaining ATP levels. This could enhance their ability to adapt to osmotic stress.

(18) Line 429-432: Conclusions about "metabolic flexibility" and "sustained energy supply" are broad. Please reframe to emphasize observed gene/protein expression trends rather than functional outcomes.

Response: We are very sorry for our ambiguous writing this sentence. This part has been revised. Lines 452-460: In summary, carbohydrate metabolism plays a pivotal role in the adaptation of Lbs. rhamnosus ATCC 53103 to osmotic stress. Enhanced glucose uptake and utilization through glycolysis and the PTS promote ATP synthesis, thereby sustaining the cellular energy supply. Meanwhile, the regulation of the pyruvate metabolic pathway optimizes the efficiency of energy utilization. Moreover, energy conservation and redox homeostasis are facilitated through the downregulation of glycogen synthesis and genes associated with the pentose phosphate pathway. Collectively, these synergistic regulatory mechanisms reinforce metabolic homeostasis and enhance the stress adaptation capacity of Lbs. rhamnosus ATCC 53103.

Comments 5.

Conclusion

The conclusion largely repeats the abstract. Consider reframing to highlight the novelty of your multi-omics findings and their limitations. For example, emphasize the observed transcriptional/proteomic shifts, while noting that physiological validation (e.g., growth, survival, metabolite assays) remains to be established.

Response: Thank you for raising this point. We have made modifications to the conclusion section. Lines 468-482: In summary, this study, for the first time, combined transcriptomic and proteomic analyses to investigate the response of Lbs. rhamnosus ATCC 53103 to osmotic stress induced by 0.6 M sodium lactate. Under these conditions, a total of 792 DEGs and 138 DEPs were identified. The differential regulation of these genes and proteins played a key role in fatty acid metabolism, amino acid metabolism and carbohydrate metabo-lism. The research results demonstrate the intrinsic connection between Lbs. rhamnosus ATCC 53103 and its metabolite synthesis regulatory network in response to osmotic pressure stress, further understanding the stress response mechanism of lactic acid bacteria exposed to osmotic pressure stress environments, and providing theoretical guidance for addressing the impact of osmotic pressure stress on the production of lactic acid bacteria products. In the future, it is necessary to explore the global response patterns under different stress intensities and combinations to reveal more universal regulatory laws. In addition, combining these molecular mechanisms with industrial fermentation processes is expected to provide new ideas for constructing highly tolerant lactic acid bacteria strains, thereby enhancing the stability and application value of lactic acid bacteria products.

Reviewer 3 Report

Comments and Suggestions for Authors

foods-3839222-peer-review-v1

The paper explore interesting topics - the stress conditions and their role on the metabolic performance. Authors have choice an appropriate model - Lacticaseibacillus rhamnosus, and species that have been explored in beneficial microbiology for decades. Strains belong to that species are well studied and applied as probiotics. Authors have explored tools of omics entourage and based on simple experimental bench work plan manage to extract much as possible information for the presence and expression of specific genes and following production of appropriate key metabolites of interests involved in studied processes.

In opinion paper deserve attention form the Editor, however, some adjustments and corrections will need to be taken into account.

Introduction in general is well presented; Material and methods will need some additional attention and some of them will need to be presented with a bit more details.

Results and Discussion are well presented in combined way and are sufficient informative and well-illustrated by the presented Figures and tables.

Ln10: Will be relevant to say that some strains belong to the species Lacticaseibacillus rhamnosus are a functional LAB... In fact, not all strains belonging to this species fits to the mentioned description. Please, consider correcting according to it.

Ln13: Since already full name of the species was introduced, in this and following occasions, please, use abbreviated version according to the recommendations from 2023: Todorov SD, Baretto Penna AL, Venema K, Holzapfel WH, Chikindas ML. Recommendations for the use of standardized abbreviations for the former Lactobacillus genera, reclassified in the year 2020. Benef Microbes. 2023 Dec 12;15(1):1-4. doi: 10.1163/18762891-20230114. PMID: 38350480.

Ln15: Explain what DEG is.

Ln31: Gram needs to with capital G, it refers to the name of Hans Gram.

Ln34: Please, abbreviation of L. rhamnosus needs to be according to previous recommendations.

Ln61: Bacillus is not LAB, is this example appropriate? Maybe somewhat this needs to be mentioned in to the text that other bacterial species, not parts of the LAB have similar or different approaches to reduce stress.

Ln65: Bacillus subtilis needs to be in italics

Ln84: Please, provide address for te mentioned university.

Please, for all material and equipment used in current study, name of the supplier accompanied by the address needs to be provided. The information will need to include name of the company, and addresses included city, state (in case of the federal country) in abbreviated way, and name of the country. In following occasions, only name of the company will be sufficient. Please, try to use name of the headquarters and not of the local distributors.

Ln92: Centrifugation needs to be as g force, and not as rpm.

Ln93: Maybe word "pellet" will be more appropriate. Word "precipitate" have different meaning.

Sentence on Ln104-107 needs to be corrected. In current way do not make sense.

Section under 2.3.1. needs to be revised and presented better.

Authors will need to change the title under 3 to Results and Discussion

Author Response

Responses to the comments of Reviewer #3

Thank you for your perfect, detailed guidance on enhancing our manuscript's readability. We've highlighted all changes and additions in the manuscript in red.

Comments 1.

Introduction in general is well presented; Material and methods will need some additional attention and some of them will need to be presented with a bit more details.

Response: Thank you for catching this. We fully agree with your opinion. The section on materials and methods requires more detailed elaboration. This is indeed an aspect that we failed to cover comprehensively enough in the manuscript. The materials and methods section has been modified as follows:

Lines 95-200:

  1. Materials and Methods

2.1 Bacterial strains and culture conditions

The Lbs. rhamnosus ATCC 53103 strain used in this study was preserved at Faculty of Food Science and Engineering, Kunming University of Science and Technology (Kunming, 650500, China). The preserved strain was inoculated in an MRS broth me-dium (hopebio, Qingdao Hope Bio-Technology Co., Ltd) at 37 ℃ for two generations as a seed solution. Previous laboratory studies indicate that a concentration of 0.6 M sodium lactate represents the sublethal concentration for Lbs. rhamnosus ATCC 53103, with late logarithmic growth occurring after 24 h [13]. Therefore, the activated Lbs. rhamnosus ATCC 53103 was inoculated with a 2 % inoculum at 37 ℃ in an MRS broth medium containing 0.6 M sodium lactate (Macklin, Shanghai Macklin Biochemical Co., Ltd) or without sodium lactate. Be recorded as SL group (the experimental group) and MRS group (the control group). After 24 h, the bacteria cells were harvested for further analysis.

2.2 Sample collection, pre-treatment and storage

For the transcriptomics analysis: collect cells during the logarithmic growth phase, then centrifuged at 8000 g for 5 min at 4 ℃. The resulting pellet was frozen in liquid nitrogen and stored at -80 ℃ until further analysis.

For the proteomics analysis: collect cells during the logarithmic growth phase. The fermentation broth was collected at the end of the logarithmic growth phase, centrifugal (as mentioned above), and the pellet was collected. The pellet was washed three times with phosphate buffered saline (PBS) (Yuanye, Shanghai yuanye Bio-Technology Co., Ltd), frozen in liquid nitrogen, and stored at -80 ℃ until further analysis.

Three biological replicates were set up for each group. All the samples were frozen and stored for subsequent detection and analysis.

2.3 Transcriptomics analysis by RNA sequencing (RNA-seq)

2.3.1 RNA extraction and preprocessing

The sample was ground in liquid nitrogen using an appropriate amount. Trizol reagent (Life Technologies, USA) was added, mixed thoroughly, and incubated for 10 min to ensure complete cell lysis. Chloroform (Guangzhou Chemical Reagent Factory) was added, mixed by inversion, and centrifuged at 14,000 × g for 10 min at 4 ℃. The mixture separated into organic and aqueous phases. The upper aqueous phase was collected, and equal volumes of chloroform and isopropanol (Guangzhou Chemical Reagent Factory) were added. Following centrifugation, the supernatant was discarded, and the pellet was washed with 75 % ethanol (Guangzhou Chemical Reagent Factory), vacuum-dried for 2 - 4 min, and resuspended in an appropriate volume of RNase free water. The solution was incubated at room temperature for 10 min to ensure complete dissolution, then mixed thoroughly, briefly centrifuged, and stored at -80 ℃.

2.3.2 Library construction, quality control, and sequencing

Total RNA was purified using Agencourt RNA Clean XP Beads (Agencourt Bio-science, USA) to deplete ribosomal RNA (rRNA). First- and second-strand cDNA syn-thesis was performed using a PCR thermal cycler. The resulting double-stranded cDNA was purified using 1.8 × Agencourt AMPure XP Beads (Agencourt Bioscience, USA), and the supernatant was collected. Adapter ligation and end repair were sub-sequently performed using a PCR thermal cycler (Dongsheng Xingye Scientific In-struments Co., Ltd., ETC811). USER enzyme was added, and the mixture was incubated at 37 ℃ for 15 min. The product was then purified using AMPure XP Beads (Agencourt Bioscience, USA), washed with 80 % ethanol, eluted with ddHâ‚‚O, and amplified by PCR, followed by a final purification. Library quality was assessed using the ABI StepOnePlus Real-Time PCR System (Life Technologies, USA). The final library was sequenced on the Illumina NovaSeq X Plus platform.

2.3.3 Analysis of differentially expressed genes in RNA-seq datasets

To compare gene expression levels across genes and samples, the dataset was normalized using the fragments per kilobase of transcript per million mapped reads (FPKM) method [14]. Gene expression levels were estimated using RSEM software and normalized using the FPKM method to eliminate the effects of gene length and sequencing depth. Differentially expressed genes (DEGs) were identified using the edgeR package (version 3.12.1), with a fold change ≥ 2 and a false discovery rate (FDR) < 0.05 as the screening threshold. Functional enrichment analysis of DEGs was performed using Gene Ontology (GO) terms and Kyoto Encyclopedia of Genes and Genomes (KEGG) pathways, with a significance threshold of q-value < 0.05.

2.4 Proteomics

2.4.1 Protein extraction and pretreatment

Samples were thawed and lysed in an appropriate volume of lysis buffer containing 1 % sodium deoxycholate and 8 M urea (both from Macklin, Shanghai Macklin Bio-chemical Co., Ltd), along with 1 × protease inhibitor to prevent protease activity. The mixture was vortexed, mixed, and homogenized three times using a high-throughput tissue mill. The resulting lysate was incubated in a sedimentation chamber at 4 ℃ for 30 min, with vortexing every 10 min. Samples were centrifuged at 14,000 × g for 20 min at 4 ℃. The resulting supernatant was collected, and protein concentration was quantified using the Pierce™ Rapid Gold BCA Protein Assay Kit (Thermo Fisher Scientific, A53225). Sample pretreatment involved protein denaturation, reduction and alkylation, enzymatic digestion, and peptide desalting. Protein pretreatment was carried out using the iST Sample Preparation Kit (PreOmics, Germany). An appropriate amount of protein was mixed with 50  µL of lysis solution. The sample was heated at 95 ℃ for 10 min at 1,000 × g. After cooling to room temperature, trypsin digestion buffer was add-ed, and the sample was incubated at 37 ℃ for 2 h with shaking at 500 × g. The enzymatic reaction was terminated by the addition of termination buffer. Peptides were desalted using the iST cartridge and eluted with two 100  µL portions of elution buffer. The eluted peptides were vacuum-dried and stored at -80 ℃.

2.4.2 DIA data acquisition

The desalted, lyophilized peptides were redissolved in a solution of phase A (0.1 % formic acid in water, Sigma-Aldrich Merck KGaA, Darmstadt, Germany) and analyzed by LC-MS/MS. The complete system consisted of a tandem UltiMate 3000 (Thermo Fisher Scientific, MA, USA) and a timsTOF Pro2 mass spectrometer (Bruker Daltonics). The samples were separated using an AUR3-15075C18 analytical column (15 cm × 75 μm i.d, 1.7 μm particle size, 120 A pore size,IonOpticks) with a 60 min gradient at a column temperature of 50 ℃. The column flow rate was controlled at 400 nL/min and the gradient started with 4 % of phase B (80 % acetonitrile and 0.1 % formic acid, Sigma-Aldrich Merck KGaA, Darmstadt, Germany) and rose to 28 % in 25 min, 44 % in 10 min, 90 % in 10 min, maintained for 7 min, and equilibrated at 4 % for 8 min. The mass spectrometer was operated in diaPASEF mode to acquire data independent acquisition (DIA) data, with a scanning range of 349-1229 m/z and an isolation window width of 40 Da. During the PASEF MS/MS scan, the collision energy increased linearly with ion mobility, rising from 59 eV (1/K0 = 1.6 Vs/cm²) to 20 eV (1/K0 = 0.6 Vs/cm²).

2.4.3 Data analysis

The Data of DIA were processed and analyzed by Spectronaut 18 (Biognosys AG, Switzerland) with default settings. Specific trypsin was set as the digestion type and digestion enzyme. Carbamidomethyl on cysteine was specified as the fixed modifica-tion. Oxidation on methionine was specified as the variable modifications. Retention time prediction type was set to dynamic iRT. Data extraction wasdetermined by Spec-tronaut based on the extensive mass calibration. Spectronaut will determine the ideal extraction window dynamically depending on iRT calibration and gradient stability. Qvalue (FDR) cutoff on precursor level was 1 % and protein level was 1 %. Decoy gen-eration was set to mutated which similar to scrambled but will only apply a random number of AA position swamps (min = 2, max = length/2). Normalization strategy was set to Local normalization. Peptides which passed the 1 % Qvalue cut off were used to calculate the major group quantities with MaxLFQ method.

Comments 2.

Results and Discussion are well presented in combined way and are sufficient informative and well-illustrated by the presented Figures and tables.

Response: Thank you for your recognition of our research results and discussion section. Your affirmation of the completeness of the chart information and the clarity of the discussion has greatly encouraged us.

Comments 3.

Ln10: Will be relevant to say that some strains belong to the species Lacticaseibacillus rhamnosus are a functional LAB... In fact, not all strains belonging to this species fits to the mentioned description. Please, consider correcting according to it.

Response: Thank you for raising this point. We have modified this sentence. Lines 12-13: Lacticaseibacillus rhamnosus is renowned for its tolerance to gastric acid and adaptability to bile and alkaline conditions, and is crucial for intestinal health and immune regulation.

Comments 4.

Ln13: Since already full name of the species was introduced, in this and following occasions, please, use abbreviated version according to the recommendations from 2023: Todorov SD, Baretto Penna AL, Venema K, Holzapfel WH, Chikindas ML. Recommendations for the use of standardized abbreviations for the former Lactobacillus genera, reclassified in the year 2020. Benef Microbes. 2023 Dec 12;15(1):1-4. doi: 10.1163/18762891-20230114. PMID: 38350480.

Response: Thank you for raising this point. We have made the necessary revisions according to the suggestions in "Todorov SD, Baretto Penna AL, Venema K, Holzapfel WH, Chikindas ML. Recommendations for the use of standardised abbreviations for the former Lactobacillus genera, reclassified in the year 2020. Benef Microbes. 2023 Dec 12;15(1):1-4. doi: 10.1163/18762891-20230114. PMID: 38350480" and have also checked the entire text. We have changed "Lacticaseibacillus rhamnosus ATCC 53103 (L. rhamnosus ATCC 53103)" to "Lacticaseibacillus rhamnosus ATCC 53103 (Lbs. rhamnosus ATCC 53103)".

Comments 5.

Ln15: Explain what DEG is.

Response: Thank you for raising this point. We have made the necessary revisions in the manuscript. We have changed "DEGs" and "DEPs" to "differentially expressed genes (DEGs)" and "differentially expressed proteins (DEPs)". Lines 13-19: In this study, integrated transcriptomic and proteomic analyses were employed to elucidate the response mechanisms of Lacticaseibacillus rhamnosus under osmotic stress, induced by exposure to 0.6 M sodium lactate, which elevates environmental osmotic pressure. It was shown that 792 differentially expressed genes (DEGs) and 138 differentially expressed proteins (DEPs) were detected in Lacticaseibacillus rhamnosus ATCC 53103 (Lbs. rhamnosus ATCC 53103) treated with osmotic stress.

Comments 6.

Ln31: Gram needs to with capital G, it refers to the name of Hans Gram.

Response: Sorry, it was our mistake. We have made the necessary revisions in the manuscript. Lines 35-36: Lactic acid bacteria (LAB) are Gram-positive bacteria that convert glucose into the final metabolite lactic acid.

Comments 7.

Ln34: Please, abbreviation of L. rhamnosus needs to be according to previous recommendations.

Response: We thank you for this comment. Based on the current norms of microbiological taxonomy, we have systematically revised the names of all microorganisms mentioned in the text to ensure that all species names comply with the terminology standards required by international academic journals. We have changed "L. rhamnosus" to "Lbs. rhamnosus". Mainly includes: Line 83, Line 96, Line 100, Line 101, Line 102, Line 201, Line 203, Line 207, Line 213, Line 217, Line 236, Line 243, Line 249, Line 255, Line 265, Line 269, Line 271, Line 276, Line 295, Line 300, Line 304, Line 307, Line 314, Line 315, Line 344, Line 356, Line 394, Line 401, Line 452, Line 459, Line 468, Line 472.

Comments 8.

Ln61: Bacillus is not LAB, is this example appropriate? Maybe somewhat this needs to be mentioned in to the text that other bacterial species, not parts of the LAB have similar or different approaches to reduce stress.

Response: Thank you for catching this. By systematically reviewing the literature cited in the introduction, we found that literature [9-11] failed to fully explain the molecular response mechanism of lactic acid bacteria under osmotic stress conditions, especially lacking in-depth data support and mechanism analysis at the levels of gene expression regulation and protein regulation. For this reason, we conducted a large-scale literature search and strict screening again, replacing and updating these three key references. The newly added literature, based on transcriptomics or proteomics research methods, has comprehensively revealed the molecular regulatory pathways of lactic acid bacteria in response to osmotic stress, including changes in key gene expression and functional regulation of related proteins, thereby providing more solid and comprehensive theoretical support for this study.

Lines 60-79: For example, under osmotic stress, the genes GshAB, GshR3, PepN, GshR4, and serA, which are associated with amino acid metabolism, are upregulated in Lactobacillus plantarum FS5-5. In addition, I526_2330, Gpd, and Gnd, which participate in carbohydrate metabolism, also show increased expression. Lactobacillus plantarum FS5-5 protects cellular proteins and macromolecules by accumulating compatible solutes, elevating GSH levels, and upregulating DNA repair proteins [9]. Another study found that under osmotic pressure stress, Lacticaseibacillus paracasei Zhang transformed into a viable but non-culturable (VBNC) state. Moreover, while maintaining cell integrity, the loss of surface smoothness and the increase in cell aggregation were observed. Further transcriptomic analysis revealed that genes related to carbohydrate and nutrient transport (such as PTS and ABC transport proteins) were upregulated, suggesting that this strain may adapt to osmotic pressure stress and maintain survival by enhancing substance uptake and metabolic capabilities [10]. Additionally, nontargeted metabolomic analysis of Bifidobacterium bifidum CCFM16 during osmotic adaptation revealed an upregulation of F6PPK, a key enzyme in the bifid shunt pathway. This finding suggests that cells redirected energy from basal metabolism toward the synthesis of osmoprotectants to mitigate osmotic stress. Under prolonged hyperosmotic stress, Bifidobacterium bifidum CCFM16 developed a protective mechanism by converting glutamic acid to proline, thereby establishing an osmoprotective system that primarily relies on proline as the adaptive solute [11].

  1. Li, M.; Wang, Q. Q.; Song, X. F.; Guo, J. J.; Wu, J. R.; Wu, R. N. iTRAQ-based proteomic analysis of responses of Lactobacillus plantarum FS5-5 to salt tolerance. Ann. Microbiol. 2019, 69, 377-394.
  2. Bao, Q. H.; Yuan, B. Y.; Ma, X. B.; Zhao, X.; Gao, R.; Jianan Li, J. N.; Kwok, L. Y. Osmotic and cold stress-induced viable but non-culturable state in Lacticaseibacillus paracasei Zhang: A transcriptome analysis. Int. Dairy J. 2025, 166, 106228.
  3. Ying Zhang, Y.; Mao, B. Y.; Tang, X.; Liu, X. M.; Zhao, J. X.; Zhang, H.; Cui, S. M.; Chen, W. Integrative genome and metabolome analysis reveal the potential mechanism of osmotic stress tolerance in Bifidobacterium bifidum. Lwt-Food Science and Technology 2022, 159, 113199.

Comments 9.

Ln65: Bacillus subtilis needs to be in italics.

Response: Thank you for raising this point. We have revised and replaced the case study and uniformly proofread all microbiological nomenclature throughout the text to ensure that all bacterial strain names adhere to the standardized italicized format. Lines 72-79: Additionally, nontargeted metabolomic analysis of Bifidobacterium bifidum CCFM16 during osmotic adaptation revealed an upregulation of F6PPK, a key enzyme in the bifid shunt pathway. This finding suggests that cells redirected energy from basal metabolism toward the synthesis of osmoprotectants to mitigate osmotic stress. Under prolonged hyperosmotic stress, Bifidobacterium bifidum CCFM16 developed a protective mechanism by converting glutamic acid to proline, thereby establishing an osmoprotective system that primarily relies on proline as the adaptive solute.

Comments 10.

Ln84: Please, provide address for te mentioned university.

Response: According to your suggestion, we have included the address of that university in the text, specifically: Kunming University of Science and Technology (Kunming, 650500, China). Lines 96-98: The Lbs. rhamnosus ATCC 53103 strain used in this study was preserved at Faculty of Food Science and Engineering, Kunming University of Science and Technology (Kunming, 650500, China).

Comments 11.

Please, for all material and equipment used in current study, name of the supplier accompanied by the address needs to be provided. The information will need to include name of the company, and addresses included city, state (in case of the federal country) in abbreviated way, and name of the country. In following occasions, only name of the company will be sufficient. Please, try to use name of the headquarters and not of the local distributors.

Response: Thank you for your attention to the methodological details. We have supplemented all the complete supplier information for the experimental materials and equipment as requested. The specific revisions include:

Lines 98-99: The preserved strain was inoculated in an MRS broth medium (Hopebio, Qingdao Hope Bio-Technology Co., Ltd) at 37 ℃ for two generations as a seed solution.

Lines 100-102: Previous laboratory studies indicate that a concentration of 0.6 M sodium lactate (Macklin, Shanghai Macklin Biochemical Co., Ltd) represents the sublethal concentration for Lbs. rhamnosus ATCC 53103, with late logarithmic growth occurring after 24 h.

Lines 112-116: The fermentation broth was collected at the end of the logarithmic growth phase, centrifugal (as mentioned above), and the pellet was collected. The pellet was washed three times with phosphate buffered saline (PBS) (Yuanye, Shanghai yuanye Bio-Technology Co., Ltd), frozen in liquid nitrogen, and stored at -80 ℃ until further analysis.

Lines 156-158: Samples were thawed and lysed in an appropriate volume of lysis buffer containing 1 % sodium deoxycholate and 8  M urea (both from Macklin, Shanghai Macklin Bio-chemical Co., Ltd), along with 1 × protease inhibitor to prevent protease activity.

Lines 174-176: The desalted, lyophilized peptides were redissolved in a solution of phase A (0.1 % formic acid in water, Sigma-Aldrich Merck KGaA, Darmstadt, Germany) and analyzed by LC-MS/MS.

Lines 180-183: The column flow rate was controlled at 400 nL/min and the gradient started with 4 % of phase B (80 % acetonitrile and 0.1 % formic acid, Sigma-Aldrich Merck KGaA, Darm-stadt, Germany) and rose to 28 % in 25 min, 44 % in 10 min, 90 % in 10 min, maintained for 7 min, and equilibrated at 4 % for 8 min.

Comments 12.

Ln92: Centrifugation needs to be as g force, and not as rpm.

Response: Sorry, it was our mistake. During the experiment, the centrifugation operation was indeed measured in terms of acceleration g. However, due to a mistake in the manuscript preparation, the unit was expressed incorrectly. We sincerely apologize for this and have conducted a systematic review and correction of the entire text to ensure the accuracy and consistency of all relevant data units.

Lines 109-111: For the transcriptomics analysis: collect cells during the logarithmic growth phase, then centrifuged at 8000 × g for 5 min at 4 ℃. The resulting pellet was frozen in liquid nitrogen and stored at -80 ℃ until further analysis.

Lines 123-124: Chloroform (Guangzhou Chemical Reagent Factory) was added, mixed by inversion, and centrifuged at 14,000 × g for 10 min at 4 ℃.

Line 161: Samples were centrifuged at 14,000 × g for 20 min at 4 ℃.

Lines 167-169: The sample was heated at 95 ℃ for 10 min at 1,000 × g. After cooling to room temperature, trypsin digestion buffer was added, and the sample was incubated at 37 ℃ for 2 h with shaking at 500 × g.

Comments 13.

Ln93: Maybe word "pellet" will be more appropriate. Word "precipitate" have different meaning.

Response: Thank you for raising this point. We have uniformly changed all occurrences of the word "precipitate" in the text to "pellet" to ensure the accuracy and consistency of the terminology usage.

Lines 109-111: For the transcriptomics analysis: collect cells during the logarithmic growth phase, then centrifuged at 8,000 × g for 5 min at 4 ℃. The resulting pellet was frozen in liquid nitrogen and stored at -80 ℃ until further analysis.

Lines 112-116: For the proteomics analysis: collect cells during the logarithmic growth phase. The fermentation broth was collected at the end of the logarithmic growth phase, centrifugal (as mentioned above), and the pellet was collected. The pellet was washed three times with phosphate buffered saline (PBS) (Yuanye, Shanghai yuanye Bio-Technology Co., Ltd), frozen in liquid nitrogen, and stored at -80 ℃ until further analysis.

Lines 127-129: Following centrifugation, the supernatant was discarded, and the pellet was washed with 75 % ethanol (Guangzhou Chemical Reagent Factory), vacuum-dried for 2 - 4 min, and resuspended in an appropriate volume of RNase free water.

Comments 14.

Sentence on Ln104-107 needs to be corrected. In current way do not make sense.

Response: Thank you for raising this point. We have restated this part of the content.

Lines 121-131: The sample was ground in liquid nitrogen using an appropriate amount. Trizol reagent (Life Technologies, USA) was added, mixed thoroughly, and incubated for 10 min to ensure complete cell lysis. Chloroform (Guangzhou Chemical Reagent Factory) was added, mixed by inversion, and centrifuged at 14,000 × g for 10 min at 4 ℃. The mixture separated into organic and aqueous phases. The upper aqueous phase was collected, and equal volumes of chloroform and isopropanol (Guangzhou Chemical Reagent Factory) were added. Following centrifugation, the supernatant was discarded, and the pellet was washed with 75 % ethanol (Guangzhou Chemical Reagent Factory), vacuum-dried for 2 - 4 min, and resuspended in an appropriate volume of RNase free water. The solution was incubated at room temperature for 10 min to ensure complete dissolution, then mixed thoroughly, briefly centrifuged, and stored at -80 ℃.

Comments 15.

Section under 2.3.1. needs to be revised and presented better.

Response: We have revised it as suggested. Thanks. We have systematically rephrased the section on transcriptomics experimental methods. Lines 119-153:

2.3 Transcriptomics analysis by RNA sequencing (RNA-seq)

2.3.1 RNA extraction and preprocessing

The sample was ground in liquid nitrogen using an appropriate amount. Trizol reagent (Life Technologies, USA) was added, mixed thoroughly, and incubated for 10 min to ensure complete cell lysis. Chloroform (Guangzhou Chemical Reagent Factory) was added, mixed by inversion, and centrifuged at 14,000 × g for 10 min at 4  ℃. The mixture separated into organic and aqueous phases. The upper aqueous phase was collected, and equal volumes of chloroform and isopropanol (Guangzhou Chemical Reagent Factory) were added. Following centrifugation, the supernatant was discarded, and the pellet was washed with 75 % ethanol (Guangzhou Chemical Reagent Factory), vacuum-dried for 2 - 4 min, and resuspended in an appropriate volume of RNase free water. The solution was incubated at room temperature for 10 min to ensure complete dissolution, then mixed thoroughly, briefly centrifuged, and stored at -80 ℃.

2.3.2 Library construction, quality control, and sequencing

Total RNA was purified using Agencourt RNA Clean XP Beads (Agencourt Bioscience, USA) to deplete ribosomal RNA (rRNA). First- and second-strand cDNA syn-thesis was performed using a PCR thermal cycler. The resulting double-stranded cDNA was purified using 1.8 × Agencourt AMPure XP Beads (Agencourt Bioscience, USA), and the supernatant was collected. Adapter ligation and end repair were subsequently performed using a PCR thermal cycler (Dongsheng Xingye Scientific Instruments Co., Ltd., ETC811). USER enzyme was added, and the mixture was incubat-ed at 37  ℃ for 15 min. The product was then purified using AMPure XP Beads (Agen-court Bioscience, USA), washed with 80 % ethanol, eluted with ddHâ‚‚O, and amplified by PCR, followed by a final purification. Library quality was assessed using the ABI StepOnePlus Real-Time PCR System (Life Technologies, USA). The final library was sequenced on the Illumina NovaSeq X Plus platform.

2.3.3 Analysis of differentially expressed genes in RNA-seq datasets

To compare gene expression levels across genes and samples, the dataset was normalized using the fragments per kilobase of transcript per million mapped reads (FPKM) method [14]. Gene expression levels were estimated using RSEM software and normalized using the FPKM method to eliminate the effects of gene length and sequencing depth. Differentially expressed genes (DEGs) were identified using the edgeR package (version 3.12.1), with a fold change ≥ 2 and a false discovery rate (FDR) < 0.05 as the screening threshold. Functional enrichment anal-ysis of DEGs was performed using Gene Ontology (GO) terms and Kyoto Encyclopedia of Genes and Genomes (KEGG) pathways, with a significance threshold of q-value < 0.05.

Comments 16.

Authors will need to change the title under 3 to Results and Discussion.

Response: Thank you for your valuable suggestions on the structure of the article. We have accordingly changed the title of the third part to "Results and Discussion".

Round 2

Reviewer 1 Report

Comments and Suggestions for Authors

Although in my opinion the manuscript still does not fully fall within the scope of foods, it is true that during the review process a lot of additional information has been added and important errors have been corrected, which has greatly improved the manuscript. Some minor questions need to be corrected prior to its publication:

“DEGs” and “DEPs” are cited only once in the abstract. Thus, it is not required the abbreviation.

Line 18: “lacticaseibacillus” should be cited in the abbreviated form the second time that is cited in the text.

Line 22: What is the meaning of “ACC” and “FAB”?

Line 35: Not only glucose, the main activity in diary foods in from lactose.

Line 77: “Bifidobacterium” should be cited abbreviated.

There is an improper space after line 92.

Line 98: “MRS” must be defined the first time that appears in the text.

Line 100-101: Manufacturer name seems to be in a different font type.

For all manufacturers, the company name, city and country must be cited the first time that appears in the text, and not “Life Technologies, USA” or Guangzhou Chemical Reagent Company).

Lines 146-153: This text seems to be in a different font type.

Line 173: Please define “DIA” and avoid use of abbreviations in headings and subheadings.

Figure 1 D and E text cannot be read. The same for Figures 3 and 4.

Insert an space after line 343. The same after lines 400 and 460.

Author Response

Responses to the comments of Reviewer #1

Comments 1.

"DEGs" and "DEPs" are cited only once in the abstract. Thus, it is not required the abbreviation.

Response: Thank you for raising this point. We have followed the suggestion and removed the abbreviations "DEGs" and "DEPs" in the abstract, using their full names instead. Lines 17-18: I It was shown that 792 differentially expressed genes and 138 differentially expressed proteins were detected in Lbs. rhamnosus ATCC 53103 treated with osmotic stress.

Comments 2.

Line 18: "lacticaseibacillus" should be cited in the abbreviated form the second time that is cited in the text.

Response: We thank you for this comment. We have revised the entire text in accordance with academic standards, ensuring that the full name and abbreviation of the term are fully indicated for its first occurrence, and that all subsequent references uniformly use the abbreviated form.

Lines 12-18: Lacticaseibacillus rhamnosus (Lbs. rhamnosus) is renowned for its tolerance to gastric acid and adaptability to bile and alkaline conditions, and is crucial for intestinal health and immune regulation. In this study, integrated transcriptomic and proteomic analyses were employed to elucidate the response mechanisms of Lbs. rhamnosus under osmotic stress, induced by exposure to 0.6 M sodium lactate, which elevates environmental osmotic pressure. It was shown that 792 differentially expressed genes and 138 differentially expressed proteins were detected in Lbs. rhamnosus ATCC 53103 treated with osmotic stress.

Lines 38-39: Lbs. rhamnosus is a species of LAB that is widely distributed in the gastrointestinal tracts of humans and animals.

Lines 82-84: However, the survival mechanism of Lbs. rhamnosus ATCC 53103 in a hyperosmotic environment is still unknown.

Comments 3.

Line 22: What is the meaning of "ACC" and "FAB"?

Response: Thank the reviewers for their attention to the terminology standards. We have provided the following supplementary explanations and revisions for the abbreviations used in the text:

ACC family: acetyl coenzyme A carboxylase.

FAB family: fatty acid-binding protein family.

Lines 18-25: Differential regulation of these genes/proteins mainly includes inhibition of fatty acid metabolism with membrane structural remodeling (down-regulation of acetyl coenzyme A carboxylase family and fatty acid-binding protein family expression), dynamic homeostasis of amino acid metabolism (restriction of synthesis of histidine, cysteine, leucine, etc., and enhancement of catabolism of lysine, tryptophan, etc.), and survival oriented reconfiguration of carbohydrate metabolism (gene expression related to the glycolytic pathway increases, while gene expression related to the pentose phosphate pathway decreases).

Comments 4.

Line 35: Not only glucose, the main activity in diary foods in from lactose.

Response: We would like to thank the reviewer for pointing out the accuracy of the content. We have revised the description in line 35 according to the suggestions. Lines 35-36: Lactic acid bacteria (LAB) are Gram-positive microorganisms that are capable of fermenting carbohydrates, producing lactic acid as the primary metabolic end product.

Comments 5.

Line 77: "Bifidobacterium" should be cited abbreviated.

Response: Thank you for raising this point. We have made the necessary revisions according to the suggestions in "Todorov SD, Baretto Penna AL, Venema K, Holzapfel WH, Chikindas ML. Recommendations for the use of standardised abbreviations for the former Lactobacillus genera, reclassified in the year 2020. Benef Microbes. 2023 Dec 12;15(1):1-4. doi: 10.1163/18762891-20230114. PMID: 38350480" and have also checked the entire text. We have changed "Bifidobacterium bifidum CCFM16" to "Bb. Bifidum CCFM16". Lines 72-79: Additionally, nontargeted metabolomic analysis of Bifidobacterium bifidum CCFM16 (Bb. Bifidum CCFM16) during osmotic adaptation revealed an upregulation of F6PPK, a key enzyme in the bifid shunt pathway. This finding suggests that cells redirected energy from basal metabolism toward the synthesis of osmoprotectants to mitigate osmotic stress. Under prolonged hyperosmotic stress, Bb. Bifidum CCFM16 developed a protective mechanism by converting glutamic acid to proline, thereby establishing an osmo-protective system that primarily relies on proline as the adaptive solute [11].

Comments 6.

There is an improper space after line 92.

Response: We are grateful for the meticulous proofreading suggestions provided by the reviewers. We have conducted a systematic review of line 92 and the entire text, removed the redundant spaces in that line, and further checked the formatting standards of the entire text (including spaces, punctuation, and line breaks) to ensure compliance with the journal requirements.

Comments 7.

Line 98: "MRS" must be defined the first time that appears in the text.

Response: We are grateful to the reviewers for their corrections regarding the terminology. We have revised the citations of "MRS" in the text according to their suggestions. Lines 97-99: The preserved strain was inoculated in an MRS (de Man, Rogosa, and Sharpe) broth medium (Qingdao Hope Bio-Technology Co., Ltd, Qingdao, China) at 37 ℃ for two generations as a seed solution.

Comments 8.

Line 100-101: Manufacturer name seems to be in a different font type.

Response: Thank you to the reviewers for their corrections regarding the formatting details. We have uniformly revised the font style of the manufacturer names in lines 100-101 to ensure that all the manufacturer names throughout the document use the same font as the main text. At the same time, we have thoroughly checked the document for similar issues and confirmed that there are no other inconsistencies. Lines 95-106: The Lbs. rhamnosus ATCC 53103 strain used in this study was preserved at Faculty of Food Science and Engineering, Kunming University of Science and Technology (Kunming, 650500, China). The preserved strain was inoculated in an MRS broth me-dium (Hopebio, Qingdao Hope Bio-Technology Co., Ltd) at 37 ℃ for two generations as a seed solution. Previous laboratory studies indicate that a concentration of 0.6 M sodium lactate (Macklin, Shanghai Macklin Biochemical Co., Ltd) represents the sublethal concentration for Lbs. rhamnosus ATCC 53103, with late logarithmic growth occurring after 24 h [13]. Therefore, the activated Lbs. rhamnosus ATCC 53103 was inoculated with a 2 % inoculum at 37 ℃ in an MRS broth medium containing 0.6 M sodium lactate (Macklin, Shanghai Macklin Biochemical Co., Ltd) or without sodium lactate. Be recorded as SL group (the experimental group) and MRS group (the control group). After 24 h, the bacteria cells were harvested for further analysis.

Comments 9.

For all manufacturers, the company name, city and country must be cited the first time that appears in the text, and not "Life Technologies, USA" or Guangzhou Chemical Reagent Company).

Response: We are grateful to the reviewers for their detailed corrections regarding the citation standards. We have uniformly revised the citation formats for all manufacturers mentioned in the text.

Lines 97-99: The preserved strain was inoculated in an MRS (de Man, Rogosa, and Sharpe) broth medium (Qingdao Hope Bio-Technology Co., Ltd, Qingdao, China) at 37 ℃ for two generations as a seed solution.

Lines 99-102: Previous laboratory studies indicate that a concentration of 0.6 M sodium lactate (Shanghai Macklin Biochemical Co., Ltd, Shanghai, China) represents the sublethal concentration for Lbs. rhamnosus ATCC 53103, with late logarithmic growth occurring after 24 h.

Lines 102-104: Therefore, the activated Lbs. rhamnosus ATCC 53103 was inoculated with a 2 % inoculum at 37 ℃ in an MRS broth medium containing 0.6 M sodium lactate (Shanghai Macklin Biochemical Co., Ltd) or without sodium lactate.

Lines 113-115: The pellet was washed three times with phosphate buffered saline (PBS) (Shanghai yuanye Bio-Technology Co., Ltd, Shanghai, China), frozen in liquid nitrogen, and stored at -80 ℃ until further analysis.

Lines 120-130: The sample was ground in liquid nitrogen using an appropriate amount. Trizol reagent (Life Technologies, USA) was added, mixed thoroughly, and incubated for 10 min to ensure complete cell lysis. Chloroform (Guangzhou chemical reagent factory, Guangzhou, China) was added, mixed by inversion, and centrifuged at 14,000 × g for 10 min at 4 ℃. The mixture separated into organic and aqueous phases. The upper aqueous phase was collected, and equal volumes of chloroform and isopropanol (Guangzhou chemical reagent factory) were added. Following centrifugation, the supernatant was discarded, and the pellet was washed with 75 % ethanol (Guangzhou chemical reagent factory), vacuum-dried for 2 - 4 min, and resuspended in an appropriate volume of RNase free water. The solution was incubated at room temperature for 10 min to ensure complete dissolution, then mixed thoroughly, briefly centrifuged, and stored at -80 ℃.

Lines 132-142: Total RNA was purified using Agencourt RNA Clean XP Beads (Agencourt Bioscience, USA) to deplete ribosomal RNA (rRNA). First and second strand cDNA synthesis was performed using a PCR thermal cycler. The resulting double-stranded cDNA was purified using 1.8 × Agencourt AMPure XP Beads (Agencourt Bioscience), and the supernatant was collected. Adapter ligation and end repair were subsequently performed using a PCR thermal cycler (Dongsheng Xingye Scientific Instruments Co., Ltd., Suzhou, China). USER enzyme was added, and the mixture was incubated at 37 ℃ for 15 min. The product was then purified using AMPure XP Beads (Agencourt Bioscience), washed with 80 % ethanol, eluted with ddHâ‚‚O, and amplified by PCR, followed by a final purification. Library quality was assessed using the ABI StepOnePlus Re-al-Time PCR System (Life Technologies). The final library was sequenced on the Illu-mina NovaSeq X Plus platform.

Comments 10.

Lines 146-153: This text seems to be in a different font type.

Response: We would like to thank the reviewers for their meticulous corrections regarding the formatting details. We have thoroughly checked the text from lines 146 to 153 as well as the entire text, and have uniformly corrected them in accordance with the journal's formatting requirements. Lines 144-152: To compare gene expression levels across genes and samples, the dataset was normalized using the fragments per kilobase of transcript per million mapped reads (FPKM) method [14]. Gene expression levels were estimated using RSEM software and normalized using the FPKM method to eliminate the effects of gene length and sequencing depth. Differentially expressed genes (DEGs) were identified using the edgeR package (version 3.12.1), with a fold change ≥ 2 and a false discovery rate (FDR) < 0.05 as the screening threshold. Functional enrichment analysis of DEGs was performed using Gene Ontology (GO) terms and Kyoto Encyclopedia of Genes and Genomes (KEGG) pathways, with a significance threshold of q-value < 0.05.

Comments 11.

Line 173: Please define “DIA” and avoid use of abbreviations in headings and subheadings.

Response: Thank the reviewers for their corrections regarding the terminology. We have revised the title.

Line 171: 2.4.2 Data independent acquisition (DIA).

Lines 181-183: The mass spectrometer was operated in diaPASEF mode to acquire DIA data, with a scanning range of 349-1229 m/z and an isolation window width of 40 Da.

Comments 12.

Figure 1 D and E text cannot be read. The same for Figures 3 and 4.

Response: Thank you for bringing this issue to our attention. We have confirmed that the quality of the images was compromised during the upload process due to compression. We have also rechecked all the charts and provided the images that met the requirements of the journal when submitting the manuscript.

Comments 13.

Insert an space after line 343. The same after lines 400 and 460.

Response: Thank you to the reviewers for their meticulous corrections regarding the layout details. We have inserted spaces at the suggested positions and thoroughly checked the formatting of the entire text.